# 🍅HUMANTOMATO: TEXT-ALIGNED WHOLE-BODY MOTION GENERATION

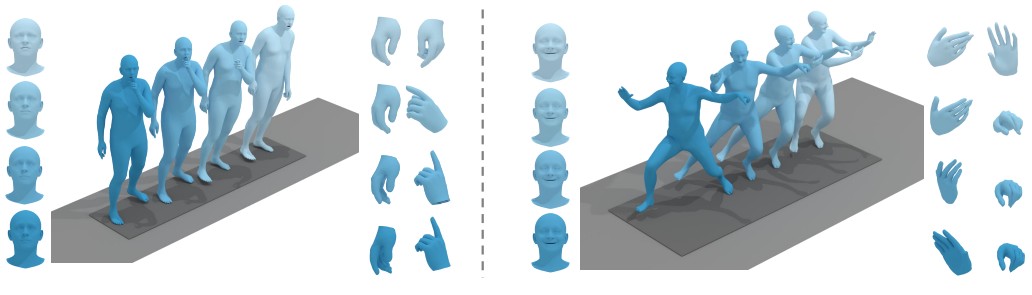

(a) stand and shush, angrily.   (b) Yang-style 40 form Tai Chi Competition routine step 34, happily.

Figure 1: The proposed HumanTOMATO can generate text-aligned whole-body motions with vivid and harmonious face, hand, and body motion. We show two generated qualitative results.

## ABSTRACT

This work targets a novel text-driven **whole-body** motion generation task, which takes a given textual description as input and aims at generating high-quality, diverse, and coherent facial expressions, hand gestures, and body motions simultaneously. Previous works on text-driven motion generation tasks mainly have two limitations: they ignore the key role of fine-grained hand and face controlling in vivid whole-body motion generation, and lack a good alignment between text and motion. To address such limitations, we propose a Text-aligned whOle-body Motion generATiOn framework, named HumanTOMATO, which is the first attempt to our knowledge towards applicable holistic motion generation in this research area. To tackle this challenging task, our solution includes two key designs: (1) a Holistic Hierarchical VQ-VAE (*aka* H²VQ) and a Hierarchical-GPT for fine-grained body and hand motion reconstruction and generation with two structured codebooks; and (2) a pre-trained text-motion-alignment model to help generated motion align with the input textual description explicitly. Comprehensive experiments verify that our model has significant advantages in both the quality of generated motions and their alignment with text. Our project codes will be public. The demo video is available in the supplementary.

## 1 INTRODUCTION

Recent years have seen an explosion of huge demand for generating high-quality 3D human motions in many scenarios, such as games, films, animation, and robotics. To reduce laborious efforts in animation creation, recent studies (Tevet et al., 2023; Chen et al., 2023b; Zhang et al., 2022; 2023a; Zhu et al., 2023) attempt to generate human motions with textual description in a natural interactive way and have achieved rapid progress in related research areas.

However, the generated motions from existing works are still unsatisfactory to meet real application needs. The problem is mainly due to two aspects. First, *existing text-driven motion generation models can only generate body-only motions rather than whole-body motions, which are highly expressive yet much more challenging.* On the one hand, the mentioned challenge comes from the limited availability of whole-body motion data. On the other hand, whole-body motion is much

more complex, where fine-grained motions of body, hand, and face should be well generated. How to model whole-body human motions is still under-explored. Second, *the generated motions lack semantic alignment with the textual description.* Existing methods adopt CLIP (Radford et al., 2021) or Large Language Models (LLMs) (Raffel et al., 2020) to provide language guidance for motion generation (Zhang et al., 2022; Tevet et al., 2023; 2022; Zhang et al., 2023c; Jiang et al., 2023). However, their alignment supervision is provided at frame level and lacks sufficient understanding of a motion at its whole sequence level. As a result, they often fail to distinguish some scenarios, such as "walking in a clockwise circle" and "walking in a counter-clockwise circle", which requires understanding motions at sequence level rather than frame level. Such a drawback severely limits the ability to generate motions well-aligned with textual descriptions.

To tackle the above issues, we propose a novel Text-aligned whOle-body Motion generATiOn framework (HumanTOMATO). The framework includes two key designs. First, *a holistic hierarchical discrete modeling strategy for body and hand motions is proposed for reconstructing and generating whole-body motions vividly.* As whole-body motion is a kind of high-dimensional spatio-temporal signal, in the first stage, we propose a Holistic Hierarchical VQ-VAE (*aka* $H^2VQ$) to compress the motion into two-level discrete codes for body and hand, respectively. In contrast, a naïve solution that simply replaces body-only motions with whole-body motions or directly increases the size of the codebook is almost in vain. The key insight of our $H^2VQ$ is learning informative and compact representations of fine-grained whole-body motions at very low bit rates. Based on the two-level discrete codes, in the second stage, we propose a Hierarchical-GPT to predict the hierarchical discrete codes of body and hand in an auto-regressive fashion. For the remaining facial motions, we generate facial expressions with a Facial conditional VAE (cVAE). Second, *a pre-trained text-motion-alignment model is introduced to enhance the textual alignment of generated motions for the first time.* For pairwise text-motion data, we pre-train a motion encoder and a text encoder, namely TMR (Petrovich et al., 2023), in a contrastive learning manner (Radford et al., 2021). Different from previous work (Zhang et al., 2022; Tevet et al., 2023; 2022; Zhang et al., 2023c; Jiang et al., 2023), we use text embedding of TMR as a language prior other than the embedding from CLIP or LLMs. In this way, the TMR provides a motion-aware language embedding for the Hirarchical-GPT to generate discrete motion codes more precisely. It is worth noting that, during training, merely supervising the prediction of discrete code tokens of body and hand is insufficient as it lacks supervision on the semantics of global motion sequence and will cause error accumulation in auto-regressive prediction. Therefore, with the text-motion similarity measured by TMR, we additionally provide text-motion alignment supervision to supervise the alignment of generated motion sequences and texts explicitly.

With these key designs, compared with previous text-driven motion generation works, HumanTOMATO can generate whole-body motions that are semantically aligned with textual descriptions, as illustrated in Figure 1. To evaluate the alignment between generated motions and input texts, we further revisit the previous retriever used for evaluating text-motion alignment and find that its retrieval ability is worse than TMR. Hence, we introduce two new criteria (*TMR-R-Precision*[(256)] and *TMR-Matching-score*), which are more accurate and challenging to evaluate the text-motion alignment in motion generation.

Before delving into details, we carefully clarify our main contributions as follows:

- To the best of our knowledge, we propose the challenging Text-driven whOle-body Motion generATiOn task for the first time and design a model (**HumanTOMATO**) to generate vivid whole-body motions that are well aligned with texts.
- To tackle the challenging whole-body motion generation problem, we introduce a $H^2VQ$ for fine-grained body and hand motion reconstruction. Accordingly, we develop a Hierarchical-GPT combined with facial cVAE to generate whole-body motions.
- To enhance the consistency and alignment between texts and motions, we pre-train text-motion-aligned encoders via a contrastive objective and introduce a sequence-level semantic supervision to help motion-text alignment.
- We propose two new criteria (*TMR-R-Precision*[(256)] and *TMR-Matching-score*), which are more accurate and challenging for evaluating text-motion alignment.

We evaluate HumanTOMATO on both whole-body (Lin et al., 2023a) and body-only (Guo et al., 2022a) motion generation benchmarks and answer four research questions based on our contributions. Experiments affirm the vividness and alignment of our generated motions with text inputs.

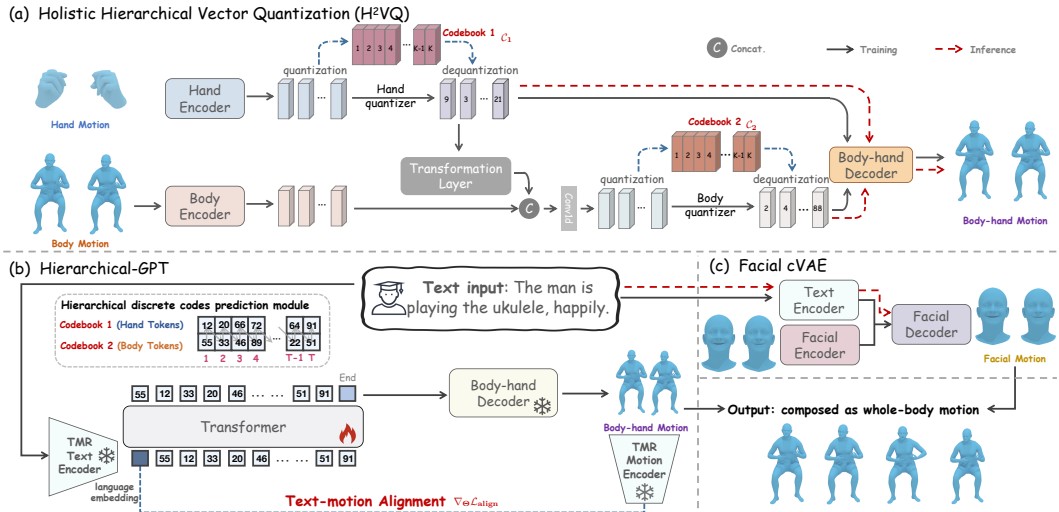

Figure 2: The framework overview of the proposed text-driven whole-body motion generation. (a) Holistic Hierarchical Vector Quantization ($H^2VQ$) to compress fine-grained body-hand motion into two discrete codebooks with hierarchical structure relations. (b) Hierarchical-GPT using motion-aware textual embedding as the input to hierarchically generate body-hand motions. (c) Facial text-conditional VAE (cVAE) to generate the corresponding facial motions. The outputs of body, hand, and face motions comprise a vivid and text-aligned whole-body motion.

## 2 METHODOLOGY

### 2.1 PROBLEM FORMULATION

We clarify notations and set up the novel research problem of text-driven **whole-body** motion generation. Given a text description $\mathbf{t}$ of a human motion, such as "*The man is playing the piano happily.*", the model should generate a vivid whole-body motion $\mathbf{m} = [\mathbf{m}_1, \mathbf{m}_2, \cdots, \mathbf{m}_L] \in \mathbb{R}^{L \times d}$ aligned with the text description, where $L$ and $d$ denote the number of frames and the dimension of the motion in each frame, respectively. As whole-body motion comes up with hand, body, and face motions, we can also decompose the $\mathbf{m}$ as $\{\mathbf{m}^H, \mathbf{m}^B, \mathbf{m}^F\}$ respectively, where $\mathbf{m}^H \in \mathbb{R}^{L \times d_h}, \mathbf{m}^B \in \mathbb{R}^{L \times d_b}, \mathbf{m}^F \in \mathbb{R}^{L \times d_f}, d = d_h + d_b + d_f$. Mathematically, we formulate the text-driven whole-body motion generation as follows:

$$\Theta^\star = \arg\max_{\Theta} P_\Theta(\mathbf{m} \mid \mathbf{t}),$$

where $\Theta$ denotes the model parameters and $P_\Theta(\cdot)$ denotes the motion distribution, respectively.

The content for the subsequent sections is outlined as follows. In Section 2.2, we first introduce $H^2VQ$ to learn fine-grained discrete motion codes for both body and hands. Then we present the Hierarchical-GPT Module in Section 2.3, which is designed to predict the text-aligned discrete motion codes for whole-body motions. Since the facial expressions are usually deterministic to textual descriptions, we adopt the method described in (Petrovich et al., 2022) to train a facial conditional VAE, thereby generating detailed expressions directly. For whole-body motion generation, we integrate motions from the body, hands, and face to produce the final output. Notably, when introducing our Hierarchical-GPT, we also explore how a text-to-whole-body motion retrieval model can benefit the text-motion alignment explicitly in Section 2.4.

### 2.2 LEARNING DISCRETE WHOLE-BODY REPRESENTATIONS

**Vanilla Motion VQ-VAE.** Motion VQ-VAE aims to learn discrete representations of human motions in an encoding-decoding fashion. Specifically, VQ-VAE recovers motions by using an auto-encoder and learns a codebook $\mathcal{C} = \{\mathbf{e}_k\}_{k=1}^K$, where $K$ denotes the codebook size and $\mathbf{e}(k)$ indicate the $k$-th embedded representation in the codebook. Given a vector $\mathbf{z}$ and the quantizer $\mathcal{Q}(\cdot; \mathcal{C})$, the quantized vector should be the element selected from the codebook $\mathcal{C}$ that can minimize the reconstruction error of $\mathbf{z}$ as,

$$\hat{\mathbf{z}} = \mathcal{Q}(\mathbf{z}; \mathcal{C}) = \arg\min_{\mathbf{e}_k} \|\mathbf{z} - \mathbf{e}_k\|_2^2.$$

In a vanilla VQ-VAE, $\mathbf{z} = \texttt{Enc}(\mathbf{m})$ indicates the latent code extracted from a motion encoder $\texttt{Enc}(\cdot)$. Thus VQ-VAE can be optimized by,

$$\mathcal{L} = \|\mathbf{m} - \texttt{Dec}(\mathcal{Q}(\mathbf{z};\mathcal{C}))\|_2^2 + \alpha\|\mathbf{z} - \texttt{sg}(\hat{\mathbf{z}})\|_2^2, \tag{1}$$

where $\alpha$ is the hyper-parameter and $\texttt{sg}(\cdot)$ is the stop-gradient operation and $\texttt{Dec}(\cdot)$ indicate the motion decoder. Different from traditional methods, the codebook $\mathcal{C}$ in motion VQ-VAE is optimized by exponential moving average ( EMA) and codebook reset (Code Reset) operations following (Razavi et al., 2019; Van Den Oord et al., 2017; Zhang et al., 2023a). Although the motivation behind discrete vector quantization in vanilla VQ-VAE is capable of compressing human motions, the quantization error present in motions can also compromise the quality of motion reconstruction, particularly for whole-body motion generation with massive details. In practice, an intuitive solution to address this would be to increase the size of the codebook. However, adopting such a scheme would evidently lead to an increase in the computational cost and quickly encounter performance bottlenecks.

**Holistic Hierarchical VQ-VAE.** Recently, the Residual Vector Quantization technique, also known as RVQ (Barnes et al., 1996; Zeghidour et al., 2021), has significantly advanced the development of music generation task (Défossez et al., 2022; Copet et al., 2023). Technically, RVQ iteratively quantizes the quantization error at each level from the previous one, reducing quantization errors effectively while maintaining a low memory cost of the codebook (see Appendix C.2 for details). Motivated by this (Défossez et al., 2022), we propose a novel Holistic Hierarchical Vector Quantization scheme, shorted by H²VQ, into the field of motion generation. Unlike RVQ, we incorporate the kinematic structure prior to the H²VQ modeling, enabling it to learn compact representations of fine-grained whole-body motions at an extremely low bit rate. Given the distinct differences in amplitude and frequency between body and hand motions, we have further designed two separate encoders and codebooks to learn discrete representations for body and hand motions.

The architecture of our proposed H²VQ is illustrated in Figure 2(a). In the encoding phase, we input hand and body motions, yielding hand and body tokens through the hand encoder $\texttt{Enc}^{\texttt{H}}(\cdot)$ and the body encoder $\texttt{Enc}^{\texttt{B}}(\cdot)$, respectively. The learned hand tokens are further quantized by the Hand Quantizer $\mathcal{Q}^H(\cdot;\mathcal{C}^H)$ as $\mathbf{z}^H$. Since the body motions are usually associated with some hand gestures, to train a more natural and coordinated body codebook, we fuse the body and hand tokens using the $\texttt{Concat}(\cdot)$ and $\texttt{Conv1d}(\cdot)$ operations. As shown in Figure 2, before this fusion, the quantized hand tokens undergo a transformation through a projection layer. After that, the fused tokens are further quantized by Body Quantizer $\mathcal{Q}^B(\cdot;\mathcal{C}^B)$ as $\mathbf{z}^B$. Finally, the hand tokens $\mathbf{z}^H$ and body tokens $\mathbf{z}^B$ are concatenated together and fed into the Body-hand Decoder to reconstruct the body-hand motions precisely.

During the training phase, the primary goal is to reconstruct motions while concurrently updating the two codebooks through the EMA and Code Reset operations. In the inference phase, after obtaining quantized code indices, the Body-Hand Decoder can generate body-hand motions by querying the respective codebooks. The algorithmic flows for these two phases can be found in Appendix C.

## 2.3 Hierarchical Whole-body Motion Generation

Given the two precise quantized codebooks of H²VQ, the motion sequence should be generated by using the corresponding decoders. The previous popular approach is to predict code indices in GPT-like auto-regressive fashion (Zhang et al., 2023a). Since the proposed H²VQ requires the use of two codebooks with structure relations, the aforementioned approach is not applicable. To better model the nature and coherence of body-hand motions, we design a hierarchical discrete codes prediction module, termed Hierarchical-GPT which is illustrated in Figure 2(b), for hand-body motions.

**Hierarchical-GPT.** The Hierarchical-GPT is built upon a transformer-based architecture, where the first input token is a textual embedding. With the input body-hand motion $\mathbf{m}^B = [\mathbf{m}_1^B, \mathbf{m}_2^B, \cdots, \mathbf{m}_L^B]$ and $\mathbf{m}^H = [\mathbf{m}_1^H, \mathbf{m}_2^H, \cdots, \mathbf{m}_L^H]$, we have corresponding code indices, denoted as $\mathbf{I}^B = [\mathbf{I}_1^B, I_2^B, \cdots, I_{L/r}^B, \texttt{End}]$ and $\mathbf{I}^H = [\mathbf{I}_1^H, I_2^H, \cdots, I_{L/r}^H, \texttt{End}]$, where 'End' indicates the end token and $r$ denotes the down-sampling rate, which is used to convert the input motion sequence to discrete motion tokens. Therefore, as shown in Figure 2(b), the code indices prediction can be

formulated as an auto-regressive prediction problem:

$$
\begin{aligned}
P(\mathbf{I}_{1,2,\cdots,L/r}^{B,H} \mid \mathbf{t}) &= \prod_{s=1}^{L/r} P(\mathbf{I}_s^{B,H} \mid \mathbf{I}_{<s}^{B,H}, \mathbf{t}) \\
&= \prod_{s=1}^{L/r} P(\mathbf{I}_s^{B} \mid \mathbf{I}_{<s}^{B,H}, \mathbf{t}) \cdot P(\mathbf{I}_s^{H} \mid \mathbf{I}_s^{B}, \mathbf{I}_{<s}^{B,H}, \mathbf{t}),
\end{aligned}
\tag{2}
$$

where we first predict the body token index and then predict the hand token index at each down-sampled timestamp $s$. As shown in Figure 2(b), the first token is the textual embedding of the given input text. Here we leverage a pre-trained text encoder to extract such an embedding. Please refer to Section 2.4 for more details. In practice, we train the prediction transformer with casual self-attention Vaswani et al. (2017). As the Hierarchical-GPT aims to predict code indices, our model is optimized with the cross-entropy loss $\mathcal{L}_{CE}$. The training details are available in Appendix B.3.

**Facial conditional VAE.** As the facial expression is partially independent of body and hand motions while highly related to the given facial descriptions and even speech, we generate the facial motion with a Facial conditional VAE (cVAE) based on given expression texts. Motivated by TEMOS (Petrovich et al., 2022), our Facial cVAE, shown in Figure 2(c), is composed of a facial encoder, a text encoder, and a facial decoder, which are optimized with the facial reconstruction loss, KL loss, and the cross-modality loss. In the inference stage, given the textual description, the text encoder and motion decoder will generate the diverse face motion according to the expression in the given text and the motion length. The training details are available in Appendix B.4.

## 2.4 Pre-trained Text-motion Retrieval Model as a Prior

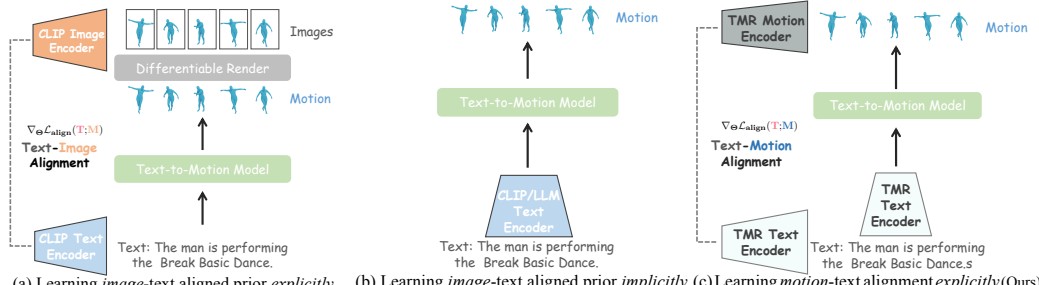

(a) Learning *image*-text aligned prior *explicitly*.  (b) Learning *image*-text aligned prior *implicitly*. (c) Learning *motion*-text alignment *explicitly* (Ours).

Figure 3: Technical comparisons on introducing language priors of existing methods.

In existing pre-trained models, there often exists a notable semantic gap between the representation of text and its corresponding motion due to the differences in the granularity of content representation between text and motion. For instance, text may involve a simple verb but its corresponding motion would require a sequence of actions. For the text-to-motion generation task, it is crucial to ensure that the textual embedding extracted from the text encoder is motion-aware. Early work primarily focused on employing various methods to narrow the representation differences between text and motion, thereby obtaining a text embedding more conducive to driving motion generation.

As shown in Figure 3(a) and Figure 3(b), we can briefly divide them into two categories. The first is *supervision by an image-text aligned prior explicitly*. As there was no strong text-motion-aligned pre-trained model, MotionCLIP (Tevet et al., 2022) renders the generated motions as images and then supervises the alignment between text and rendered images with the CLIP model. However, the image encoder of CLIP is a strong supervision of static image content understanding, which is quite different from dynamic motion. This supervision will cause the generated motion to be *over-smoothing*, even *stillness* (see Appendix E). Accordingly, supervising generated motion via a text-image-aligned prior is inappropriate. The second is *learning with image-text aligned prior implicitly*. Existing attempts (Tevet et al., 2023; Zhang et al., 2023a; 2022; Zhao et al., 2023; Yuan et al., 2023) take the CLIP text embedding as the language prior to the text-motion model training. On the one hand, it learns the motion-text alignment implicitly with pairwise data, and there is no supervision to discriminate whether the generated motion is aligned with the text explicitly. On the other hand, the CLIP text embedding is aligned with visual content, lacking the understanding of dynamic motion clues, which cannot provide sufficient spatial-temporal information to generate text-aligned motion.

Therefore, it is essential to introduce a text-motion-aligned pre-training method, ensuring that the trained text encoder can output textual embeddings more conducive to accomplishing text-to-motion generation tasks, instead of adapting from the image-text-aligned model.

Motivated by Petrovich et al. (2023), we pre-train a motion encoder and a text encoder via aligning Text and Motion in a contrastive way (Radford et al., 2021) through a Retrieval target, named TMR. Different from previous work (Zhang et al., 2022; Tevet et al., 2023; 2022; Zhang et al., 2023c; Jiang et al., 2023), the text embedding of TMR plays the role of motion-aware language prior other than the embedding from CLIP or LLMs, which is beneficial for generating text-aligned motions. The TMR is trained in a contrastive way (Radford et al., 2021) to align the motion and text embeddings, as illustrated in Figure 3(c) We leave training details in Appendix D.

Based on the pre-trained TMR, we explore enhancing the alignment between given text and generated motions from two aspects. The first is *replacing the CLIP text encoder with the TMR encoder*. Compared with the CLIP text encoder, the pre-trained TMR text encoder provides text embeddings aligned better with dynamic human motions. With the motion-aware language prior, our model can capture motion sequentiality, directions, and dynamics better than text-image-aligned language prior. The second is *introducing the motion-text alignment supervision with TMR*. When training, we feed the generated motion and the given text into the pre-trained TMR motion encoder and text encoder, respectively, to obtain both motion and text embeddings. Then, we calculate a contrastive loss $\mathcal{L}_{align}$ (Radford et al., 2021) for supervising the motion-text alignment. Accordingly, the weighted contrastive loss $\eta\mathcal{L}_{align}$ is added to the optimization objective, where $\eta$ is the hyper-parameter. The proposed optimization objective provides explicit supervision for text-motion alignment.

## 2.5 Model Training and Inference

**Model Training.** In the first stage, similar to the vanilla VQ (Eqn. 1), H$^2$VQ is optimized by:

$$\mathcal{L} = \|\mathbf{m} - \text{Dec}\left(\mathcal{Q}^H(\mathbf{z}^H; \mathcal{C}^H), \mathcal{Q}^B(\mathbf{z}^B; \mathcal{C}^B)\right)\|_2^2 + \alpha\left(\|\mathbf{z}^H - \text{sg}(\hat{\mathbf{z}}^H)\|_2^2 + \|\mathbf{z}^B - \text{sg}(\hat{\mathbf{z}}^B)\|_2^2\right). \quad (3)$$

Besides, the codebooks are optimized by EMA and Code ReSet techniques. In the second stage, we train the Hierarchical-GPT with both the cross-entropy loss $\mathcal{L}_{CE}$ and the text-motion alignment loss $\mathcal{L}_{align}$, overall as $\mathcal{L}_{CE} + \eta\mathcal{L}_{align}$.

**Model Inference.** In the inference phase, we first extract the text embedding from TMR. Then we input the TMR textual embedding as the initial token into the Hierarchical-GPT, which then predicts discrete body and hand tokens in an auto-regressive manner. The body and hand tokens are fed into the Body-hand Decoder to generate text-aligned human motion. Ultimately, incorporating the facial motions produced by the facial cVAE, we output the comprehensive whole-body motions.

## 3 Experiments

In this section, we evaluate the proposed HumanTOMATO model on both whole-body and body-only motion generation benchmarks. Besides, we will also present ablations on each technical design of our method. We structure the experiments to answer the following **four** research questions (RQs).

- **RQ1:** Does our proposed HumanTOMATO model outperform existing generation methods on the whole-body motion generation task?
- **RQ2:** How do hierarchical discrete representations of whole-body motions help improve the quality of motion generation?
- **RQ3:** How does the pre-trained text-motion retrieval model help the alignment between the generated motions and texts?
- **RQ4:** Why are the proposed evaluation metrics on alignment between generated motions and given texts more accurate and challenging?

## 3.1 Datasets and Evaluation

### 3.1.1 Whole-body and Body-only Datasets

**Motion-X** (Lin et al., 2023a) is currently the largest 3D whole-body motion-text dataset, which consists of 95,642 high-quality human motions along with 95,642 text captions. In Motion-X,

GRAB (Taheri et al., 2020) is a representative subset with vivid grab motions, which is used for our ablation study.

**HumanML3D** (Guo et al., 2022a) is currently the largest 3D body-only motion-text dataset, which consists of 14,616 high-quality human motions along with 44,970 text captions.

We take the Motion-X dataset to evaluate the whole-body motion generation task and take the HumanML3D dataset to perform ablations and verify the generalizability of our proposed solution to the body-only motion generation setting. We follow (Lin et al., 2023a; Guo et al., 2022a) to split these datasets into training, validation, and test sets with proportions of $80\%$, $5\%$, and $15\%$.

### 3.1.2 EVALUATION DETAILS

We quantitatively evaluate the generated motions from three aspects. (1) **The quality of the generated motions.** *Frechet Inception Distance (FID)* is adopted to measure the gap between the distributions of the generated and real motions. (2) **The alignment between texts and generated motions.** *Matching-score* is used to measure the similarity between texts and the generated motions and *R-Precision*$^{(B)}$ is used to measure the motion-to-text retrieval accuracy in a $B$-size retrieval pairwise motion-text set. (3) **Generation diversity.** We use *Diversity* to evaluate the average extracted feature Euclidean distances among 300 randomly sampled motion pairs and use *MModality* to measure the generation diversity within the same given text.

To better evaluate the alignment between generated motions and texts, we additionally introduce new metrics to evaluate text-motion alignment from two aspects: (1) **More accurate evaluation.** Previous works used the feature extractor from Guo et al. (2022a) to calculate the *R-Precision*$^{(B)}$ and *Matching-score*. However, its retrieval accuracy is not as accurate as the TMR described in Section 2.4 (comparison in Section 3.6). Therefore, we introduce *TMR-R-Precision*$^{(B)}$ and *TMR-Matching-score* to evaluate the text-motion alignment, where the feature extractor is replaced by TMR but not the retriever in Guo et al. (2022a). (2) **More challenging evaluation metrics.** Retrieval of corresponding texts in a 32-size set is easier than retrieval in a larger size set. Therefore, we add a new retrieval setting as $B = 256$. The comparison between these two settings will be shown in Section 3.6. Besides, we also provide qualitative comparisons of visualization results.

We compare our HumanTOMATO with existing state-of-the-art baselines. (1) **T2M-GPT**: The T2M-GPT (Zhang et al., 2023a) method learns discrete representations for motions at first, and then introduces a GPT-like codes prediction mechanism in the second stage with CLIP prior. (2) **MDM**: MDM (Tevet et al., 2023) is a pioneering work that introduces diffusion models into the field of action generation. (3) **MLD**: Motivated by latent diffusion models (Rombach et al., 2022), MLD (Chen et al., 2023b) learns motion latent representations for motion VAE via a diffusion model.

### 3.2 IMPLEMENTATION DETAILS

**Motion representation.** For body-hand motion representations, inspired by the motion representation (H3D-Format) in (Guo et al., 2022a), we expand the body-only representation to holistic body-hand motion representation. Specifically, the $i$-th pose is defined by a tuple of root angular velocity $\dot{r}^a \in \mathbb{R}$ along the Y-axis, root linear velocities ($\dot{r}^x, \dot{r}^z \in \mathbb{R}$) on XZ-plane, root height $r^y \in \mathbb{R}$, local joints positions $\mathbf{j}^p \in \mathbb{R}^{3N-1}$, and velocities $\mathbf{j}^v \in \mathbb{R}^{3N}$, where $N$ denotes the number of whole body joints, including both body joints and hand joints. For face motion representations, we follow the Flame Format (Kim et al., 2023) and use $\mathbf{f} \in \mathbb{R}^{50}$ to represent the face expression. Thus, we represent the whole-body motion as $x^i = \{\mathbf{f}, \dot{r}^a, \dot{r}^x, \dot{r}^z, \dot{r}^y, \mathbf{j}^p, \mathbf{j}^v\}$. We conduct a set of ablation studies on HumanML3D based on VAE and VQ-VAE to justify the motion format. Please refer to Appendix B.1 for more details.

**Training details.** All our experiments are trained with the AdamW (Loshchilov & Hutter, 2019) optimizer using a fixed learning rate of $10^{-4}$ on four NVIDIA Tesla A100-80GB GPUs and are tested on one NVIDIA Tesla A100-80GB GPU. Training batch size is set to 256 for both H$^2$VQ and Hierarchical-GPT stages. Each experiment is trained for 6,000 epochs during H$^2$VQ stages and 2,000 epochs during Hierarchical-GPT stages. Please refer to Appendix B for more details.

| | FID↓ | R-Precision[(32)] | | | TMR-R-Precision[(256)] | | | TMR-Matching Score ↓ | Matching Score ↓ | MModality↑ | Diversity↑ |
|---|---|---|---|---|---|---|---|---|---|---|---|
| | | Top1↑ | Top2↑ | Top3↑ | Top1↑ | Top2↑ | Top3↑ | | | | |
| GT | - | 0.500 | 0.708 | 0.814 | 0.407 | 0.578 | 0.673 | 0.768 | 2.888 | - | 11.087 |
| T2M-GPT | 1.366 | 0.368 | 0.553 | 0.655 | 0.310 | 0.446 | 0.527 | 0.881 | 4.316 | 2.356 | 10.753 |
| MDM | 3.800 | 0.352 | 0.547 | 0.6341 | 0.310 | 0.430 | 0.530 | 0.840 | 4.050 | **2.530** | **11.400** |
| MLD | 3.407 | 0.385 | 0.571 | 0.683 | 0.333 | 0.477 | 0.561 | 0.883 | 3.901 | 2.448 | 10.420 |
| HumanTOMATO | **1.174** | **0.416** | **0.603** | **0.703** | **0.399** | **0.555** | **0.638** | **0.809** | **3.894** | 1.732 | 10.812 |

Table 1: Main results of motion generation on Motion-X dataset.

### 3.3 MAIN RESULTS ANALYSIS (RQ1)

We answer RQ1 from both quantitative and qualitative aspects. (1) *Quantitative results.* We quantitatively compare our method with baselines from generation quality, text-motion alignment, and diversity, which are shown in Table 1. The metrics show that our method wins baseline methods on generation quality and text-motion alignment. The mean values are reported in Table 1. The standard value is reported in Appendix F. (2) *Qualitative results.* We compare our method with MLD and T2M-GPT. The comparison results show that our method has a stronger ability on generation quality (hand and body). For the "Flying Kick" case, MLD and T2M-GPT fail to generate desirable motions, but our method achieves it. For the second case, MLD fails to generate "forward" motion, and motions generated by T2M-GPT walk backward first and finally walk forward. In contrast, HumanTOMATO generates a vivid motion corresponding to textual descriptions.

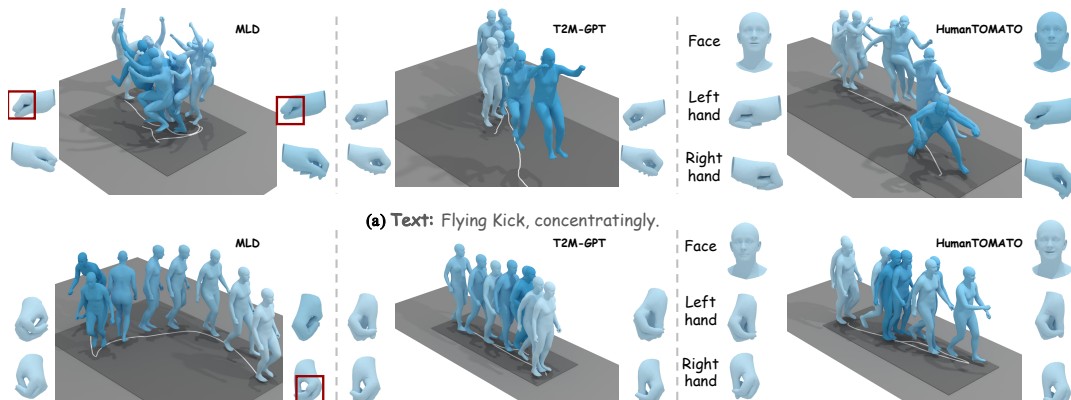

(a) **Text:** Flying Kick, concentratingly.

(b) **Text:** a person walks forwards, then suddenly, as if bumping into something, starts walking backwards, fearfully.

Figure 4: Qualitative comparison with baselines. HumanTOMATO supports face motion generation and outperforms MLD and T2M-GPT on hand motion generation and text-motion alignment.

### 3.4 ABALTION ON HIERARCHICAL DISCRETE REPRESENTATIONS (RQ2)

We compare the reconstruction result of our $H^2VQ$ with the Vanilla VQ (512 or 1024 codebook size) and RVQ methods on Motion-X and GRAB (Motion-X subset), which are shown in Table 2. Following the commonly used evaluation protocols on motion reconstruction, we take the MPJPE metric (Gower, 1975; Lin et al., 2023a; Chen et al., 2023b) to evaluate the reconstruction performance. As shown in Table 2, increasing the codebook size naïvely is almost in vain. The hierarchical modeling strategy improves the reconstruction performance significantly when learning informative low-dimensional representations. Moreover, our $H^2VQ$ is better than RVQ in reconstructing whole-body motions, with gains mainly coming from the modeling of body and hand discrete codes explicitly. When verifying the key insight of our hierarchical modeling on body-only datasets, in contrast to HumanML3D only including body-part motions, we compare the Vinilla VQ-VAE with the RVQ technique to verify our motivation. The results are shown in Appendix G. More evaluation metrics on PA-MPJPE and Acceleration error (Accel.) are also available in Appendix G.

### 3.5 TEXT-MOTION RETRIEVAL MODEL AS A PRIOR (RQ3)

Here, we evaluate our core design of introducing a pre-trained text-motion retrieval model as a prior. Ablation results in Table 3 shows that our introduced motion-aware prior benefits the alignment

| | Motion-X | | | GRAB | | | HumanML3D |
|---|---|---|---|---|---|---|---|
| | All↓ | Body ↓ | Hand↓ | All ↓ | Body ↓ | Hand ↓ | Body ↓ |
| Vanilla VQ (512) | 140.66 | 92.197 | 46.4517 | 78.228 | 38.285 | 31.480 | 77.209 |
| Vanilla VQ (1024) | 139.33 | 91.768 | 46.399 | 76.013 | 37.340 | 29.887 | 71.335 |
| RVQ | 110.94 | 73.974 | 40.011 | 62.938 | 31.124 | 27.283 | **63.051** |
| H$^2$VQ | **92.966** | **62.34** | **37.1957** | **46.742** | **24.327** | **24.588** | - |

Table 2: Comparison of the motion reconstruction errors (MPJPE in mm) of different quantization methods on Motion-X, GRAB, and HumanML3D. Our H$^2$VQ shows significant improvements.

between the generated motions and text. Visualization results in Appendix H show that our key design helps capture the motion dynamic clues better on sequentiality, directions, and dynamics.

| embedding | supervision | FID ↓ | R-Precision$^{(32)}$ | | | TMR-R-Precision$^{(256)}$ | | | TMR-Matching-score ↓ | Matching-score ↓ |
|---|---|---|---|---|---|---|---|---|---|---|
| | | | Top1 ↑ | Top2 ↑ | Top3 ↑ | Top1 ↑ | Top2 ↑ | Top3 ↑ | | |
| GT | | 0.002 | 0.500 | 0.708 | 0.814 | 0.407 | 0.578 | 0.673 | 0.768 | 2.888 |
| CLIP | ✗ | **1.086** | 0.405 | 0.588 | 0.695 | 0.345 | 0.490 | 0.573 | 0.844 | 3.917 |
| TMR | ✗ | 1.290 | 0.416 | 0.596 | 0.699 | 0.395 | 0.550 | 0.637 | 0.815 | 3.950 |
| TMR | ✔ | 1.174 | **0.416** | **0.603** | **0.703** | **0.399** | **0.555** | **0.638** | **0.809** | **3.894** |

Table 3: Abaltion on pre-trained text-motion-aligned model for motion generation on Motion-X. Both TMR embedding and text-motion alignment supervision help generate text-aligned motions.

### 3.6 Analysis on Evaluation Metrics (RQ4)

We answer RQ4 from two aspects. (1) **Our TMR-R-Precision$^{(B)}$ and TMR-Matching-score$^{(B)}$ metrics are more accurate than the R-Precision$^{(B)}$ and Matching-score metrics (Guo et al., 2022a)**. As shown in Figure 5, our TMR (in blue) shows stronger retrieval ability than Guo et al. (2022a)'s retriever (in red) on both $B = 32$ and $B = 256$ settings. Moreover, Guo et al. (2022a)'s retriever shows a larger retrieval gap than TMR when changing $B = 32$ to $B = 256$. Therefore, TMR can evaluate text-motion alignment more accurately than (Guo et al., 2022a). (2) $B = 256$ **is a more challenging retrieval setting than the $B = 32$ setting.** Retrieving text from motion in 32 text candidates is much easier than 256 candidates.

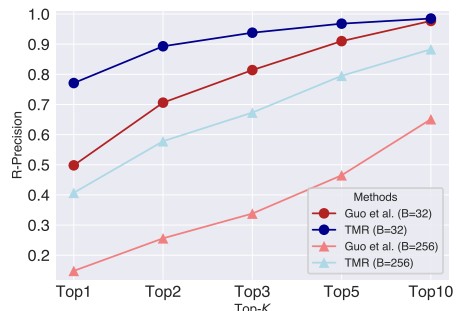

Figure 5: Comparison with existing metrics on Motion-X. Existing evaluation metrics Guo et al. (2022a) are illustrated in red, and ours are in green. The $B = 32$ and $B = 256$ settings for retrieval are denoted as "—●—" and "—▲—" respectively.

As shown in Figure 5, when changing $B = 32$ to $B = 256$, the retrieval accuracy of both retrievers declines due to the increased size of the retrieval set, which makes the evaluation protocols more challenging. Overall, with a higher upper limit of retrieval capabilities, *TMR-R-Precision$^{(256)}$* can better evaluate the performance of different methods on text-motion alignment. Additionally, *TMR-R-Precision$^{(B)}$* and *TMR-Matching-score* are also more accurate and challenging metrics on the body-only dataset (HumanML3D). More details and numerical comparisons are in Appendix I.

## 4 Conclusion

This work studies the problem of text-driven whole-body motion generation. We carefully clarify the existing challenges in generating vivid text-aligned whole-body motion on motion reconstruction and text-motion alignment. To tackle the challenges, two main technical contributions are proposed: (1) a Holistic Hierarchical VQ-VAE (H$^2$VQ) and a Hierarchical-GPT for fine-grained body and hand motion reconstruction and generation, and (2) a pre-trained text-motion-alignment model to help generate text-aligned motion. We conduct comprehensive experiments to verify the superiority and effectiveness of the proposed solution on Motion-X and HumanML3D datasets. Experimental results show that HumanTOMATO can generate vivid text-aligned whole-body motion. The broader impact and limitation are discussed in Appendix K.

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

# Appendix:
# 🍅HumanTOMATO: Text-aligned Whole-body Motion Generation

## A  RELATED WORK

**Human Motion Generation:** Generating human motions can be divided into two main types according to inputs: motion synthesis (1) without any conditions (Yan et al., 2019; Zhao et al., 2020; Zhang et al., 2020; Cai et al., 2021) and (2) with some given conditions, such as text, audio, music, and interactive scenes (Ahn et al., 2018; Petrovich et al., 2022; Zhao et al., 2023; Zhang et al., 2022; Chen et al., 2023b; Guo et al., 2022a; Ahuja & Morency, 2019; Ghosh et al., 2021; Zhang et al., 2023a; Lee et al., 2023; Yi et al., 2023b; Wang et al., 2022a; Zhou & Wang, 2023; Wang et al., 2022b; Guo et al., 2022b; Xu et al., 2022; Tseng et al., 2023; Siyao et al., 2022; Liu et al., 2022; Zhu et al., 2023; Xu et al., 2023b; Dabral et al., 2023). The second type will be more challenging and applicable due to either extracting and understanding motion and conditions or cross-modality alignment. To generate diverse, natural, high-quality human motions, many generative models have been explored by (Wang et al., 2020; Yu et al., 2020; Guo et al., 2020; Zhang et al., 2023a). Recently, diffusion-based models significantly improved the motion generation performance and diversity (Chen et al., 2023b; Tevet et al., 2023; Zhang et al., 2022; Chen et al., 2023a; Xu et al., 2023a) with stable training. However, as human motion is a kind of high-dimensional spatio-temporal signal, these methods are still hard to tackle the motion data easily. Guo et al. (2022b); Tevet et al. (2023); Chen et al. (2023b); Zhang et al. (2023a) learn low-dimensional motion latent in an encoding-decoding fashion, like VAE and VQ-VAE, in the first stage. Then, text-aligned motion latent representations could be easier to learn in the second stage. For holistic human motion generation with facial expressions and hand gestures, co-speech expression generation and gesture generation from human speech is also an arising topic in this area (Habibie et al., 2021; Yi et al., 2023a). Specifically, TalkSHOW (Yi et al., 2023a) takes the first attempt for face, hand, and body motion modeling via separate models since the facial expressions (*e.g.*, lip movement) are strongly correlated with the speech signals, but the body and gesture motions are many-to-many mappings. Bearing the difficulties in jointly modeling the whole-body motions and the lack of whole-body data, there are no existing methods to explore text-driven whole-body motion generation.

**Text-driven Motion Generation:** Text plays an important role in controlling human motion generation since it can describe the actions, directions, and dynamic body-part clues via a natural interaction way. Based on existing action recognition and motion capture datasets (Plappert et al., 2016; Mahmood et al., 2019; Liu et al., 2019; Punnakkal et al., 2021; Guo et al., 2022a), text-driven motion generation has achieved rapid progress in recent years. The input text went from the original single-action category to multiple actions and arbitrary natural language (Ahn et al., 2018; Lee et al., 2023; Lu et al., 2022; Petrovich et al., 2022; 2021; Kim et al., 2023; Zhang et al., 2023b). The generated motions also range from upper-body motions to full-body motions (additionally with global trajectories and lower-body motions) and from short-time actions to long-term motions (Plappert et al., 2018; Ahuja & Morency, 2019; Chen et al., 2023b; Zhang et al., 2023a; Athanasiou et al., 2022). Early attempts (Tevet et al., 2022; Guo et al., 2022a) heavily rely on the given motion-text datasets, making the generated motion hard to generalize. For open-vocabulary motion generation, some works try to introduce large-scale pre-trained models (e.g., CLIP (Radford et al., 2021), and LLMs (Jiao et al., 2023; Floridi & Chiriatti, 2020)) to make the text encoding powerful (Lucas et al., 2022; Jiang et al., 2023; Zhang et al., 2023c; Tevet et al., 2022; Hong et al., 2022; Lin et al., 2023c). However, existing methods suffer from two main issues. First, text-driven holistic motion generation is under-explored, while coherent hand gestures and facial expressions are essential to whole-body motions. Second, the distribution of motion is quite different from images, making CLIP prior weak in text-motion alignment, while LLMs only have textual priors. That is to say, previous efforts have not thoroughly explored motion-text alignment. Accordingly, modeling whole-body motion and exploring how to use motion-text-aligned priors are urgent for the community.

## B  IMPLEMENTATION DETAILS

### B.1  MOTION REPRESENTATION

The raw motion representation consists of two parts (Aberman et al., 2020), static part (joints offsets) and dynamic part (joint movements) respectively. We further define motion generation tasks as generating diverse and vivid joint movements based on a uniform skeleton. We follow Guo et al. (2022a) (*i.e.* H3D-Format) to randomly select a skeleton as a target skeleton, including body and hand joints, and retarget each motion sequence to it. As all motions share the same skeleton, in this way, we set the local joint offsets for all motions to be unchanged. As a pose can be decomposed as twist and swing (Li et al., 2021), vanilla inverse kinematic (IK) algorithms will ignore the twist rotation, which will lead to the wrong supervision of joint movements. To verify whether rotation regularization helps motion generation and reconstruction, we take motion reconstruction as a pretext task. For motion reconstruction, we take a transformer-based VAE Chen et al. (2023b) and convolution-based VQ-VAE (Zhang et al., 2023a) as the architecture to evaluate the motion reconstruction performance on HumanML3D. As shown in Table 4, motion without rotation information reduces the reconstruction error. Besides, the results in Table 4 show that velocity is beneficial to motion reconstruction.

As discussed in the main paper (Section 3.2), for body-hand motion representations, we take the H3D-Format (Guo et al., 2022a) as a basis and expand the body-only representation to holistic body-hand motion representation. Specifically, the *i-th* frame pose is defined by a tuple of root angular velocity ($\dot{r}^a \in \mathbb{R}$) along Y-axis, root linear velocities ($\dot{r}^x, \dot{r}^z \in \mathbb{R}$) on XZ-plane, root height $r^y \in \mathbb{R}$, local joints positions ($\mathbf{j}^p \in \mathbb{R}^{3N-1}$), and velocities ($\mathbf{j}^v \in \mathbb{R}^{3N}$), where $N$ denotes the number of joints. For face motion representations, we follow Flame Format (Kim et al., 2023) and use $\mathbf{f} \in \mathbb{R}^{50}$ to represent the face expression. Thus, we represent the whole-body motion at frame $i$ as $\mathbf{m}_i = \{\mathbf{f}, \dot{r}^a, \dot{r}^x, \dot{r}^z, \dot{r}^y, \mathbf{j}^p, \mathbf{j}^v\}$.

|  | Input Format | MPJPE | PA-MPJPE | ACCEL. |
|---|---|---|---|---|
|  | H3D-format | 49.51 | 39.47 | 7.131 |
| VAE | w/o rotation, w/o velocity | 51.43 | 39.47 | 7.27 |
|  | w/o rotation | **46.84** | **36.16** | **6.603** |
|  | H3D-format | 78.24 | 45.77 | 8.757 |
| VQ-VAE | w/o rotation, w/o velocity | 76.34 | 39.98 | 8.622 |
|  | w/o rotation | **68.86** | **39.97** | **8.274** |

Table 4: Ablation study of different motion representations on the Humanml3D dataset.

### B.2  IMPLEMENTATION DETAILS OF HIERARCHICAL MOTION VQ-VAE

We take Conv1d($\cdot$) with skip connection as the basic module for both the body encoder and the hand encoder and downsample the feature from the body part by $2\times$ and the feature from the hand part by $4\times$, respectively. In detail, at each down-sampled timestamp, the number of body tokens is 2, and the number of hand tokens is 4. Therefore, we predict two body tokens at each down-sampled timestamp. The codebook size for both hand quantizer and body quantizer is set to $512 \times 512$. That is to say, $K = 512$ and the dimension of each code is 512. We take the AdamW (Loshchilov & Hutter, 2019) as an optimizer with a fixed learning rate $1 \times 10^{-4}$, batch size of 256, and exponential moving constant $\lambda = 0.99$. The $\alpha$ in H$^2$VQ loss $\mathcal{L}$ (Eqn. 3) is set as 0.02. The body-hand decoder upsamples the feature by $2\times$. All upsampling operation in the decoder is the nearest upsampling with a scaling factor of $2\times$. Training the H$^2$VQ takes about 8 hours on 4 Tesla A100-80GB GPUs.

### B.3  IMPLEMENTATION DETAILS OF THE HIERARCHICAL-GPT.

We employ 18 transformer layers with a dimension of 1024 and 16 heads. Since the design of different downsampling rates between two codebooks, we simply concat the tokens from pre-trained H$^2$VQ stage and set the maximum length of the code index sequence as 149.

**Training.** We combine the tokens from the hand codebook $\mathcal{C}_1$ and the body codebook $\mathcal{C}_2$ and feed them into the transformer, in which we employ a causal mask with $mask_{i,j} = -\infty \times \mathbf{1}(i <$

$j) + \mathbf{1}(i \geq j)$, where $\mathbf{1}(\cdot)$ is the indicator function, to prevent information leakage from the following tokens. We employ the CLIP-ViT-L-14 model and pre-trained TMR Text encoder as the text encoder to encode the text, respectively, and freeze them in training. All the trainings are conducted on 4 Tesla A100-80GB GPUs, and cost 60 hours.

**Inference.** When performing inference, we feed the text encoder with raw texts and get the text embedding. Our Hierarchical-GPT predicts motion tokens in an autoregressive fashion with the start token of text embedding. All our tests and inferences are conducted on a single Tesla A100-80GB GPU.

### B.4 FACIAL CVAE

As discussed in the main paper (Section 2.3), we take the cVAE (Petrovich et al., 2022) to generate text-driven facial motions. The facial VAE consists of three components. (1) A facial encoder. The Facial is a 6-layer transformer. The input facial motion is concatenated with a $\mu_F$ token and a $\Sigma_F$ token. (2) A text encoder. The Facial is composed of a pre-trained DistllBERT Sanh et al. (2019) and a 6-layer transformer. The input DistillBERT feature is concatenated with a $\mu_T$ token and a $\Sigma_T$ token. (3) A facial decoder. The 6-layer transformer-based facial decoder generates facial motions from the $z_F$ or $z_T$ vector, which can be sampled from Gaussian distribution $\mathcal{N}(\mu_F, \Sigma_F)$ or $\mathcal{N}(\mu_T, \Sigma_T)$ via re-parameterizing trick (Kingma & Welling, 2013). The training loss consists of three components. (1) facial motion reconstruction loss:

$$\mathcal{L}_{rec} = \texttt{SmoothL1}(\mathbf{m}^F, \hat{\mathbf{m}}^F),$$

where $\mathbf{m}^F, \hat{\mathbf{m}}^F$ are facial motions and reconstructed facial motions and $\texttt{SmoothL1}(\cdot)$ is the SmoothL1-Loss. (2) KL Loss:

$$\begin{aligned}\mathcal{L}_{KL} =& \texttt{KL}(\mathcal{N}(\mu_F, \Sigma_F), \mathcal{N}(\mathbf{0}, \mathbf{I})) + \texttt{KL}(\mathcal{N}(\mu_T, \Sigma_T), \mathcal{N}(\mathbf{0}, \mathbf{I})) \\ &+ \texttt{KL}(\mathcal{N}(\mu_F, \Sigma_F), \mathcal{N}(\mu_T, \Sigma_T)) + \texttt{KL}(\mathcal{N}(\mu_T, \Sigma_T), \mathcal{N}(\mu_F, \Sigma_F)),\end{aligned}$$

where $\texttt{KL}(\cdot)$ is the Kullback-Leibler divergence function and $\mathcal{N}(\mathbf{0}, \mathbf{I})$ is the Gaussian distribution. (3) Cross-modal embedding similarity loss:

$$\mathcal{L}_E = \texttt{SmoothL1}(z_F, z_T).$$

The overall training loss is $\mathcal{L} = \mathcal{L}_{rec} + \lambda_1 \mathcal{L}_{KL} + \lambda_2 \mathcal{L}_E$, where $\lambda_1 = \lambda_2 = 1 \times 10^{-5}$. In the inference stage, the text encoder encodes text embedding $z_T$ first, and then feeds it into the facial decoder to obtain the facial motions.

## C Algorithm Flow of H²VQ and comparison with Residual Vector Quantization

### C.1 Training and Inference of H²VQ

In the main paper, we introduce the training and inference details in Section 2.2. For reading convenience, we provide the training and inference procedure of our Holistic Hierarchical VQ-VAE (H²VQ-VAE) in Algorithm 1 and Algorithm 2.

---

**Algorithm 1:** Training procedure of Holistic Hierarchical VQ-VAE (H²VQ-VAE)

---

**Input:** The initialized hand codebook $\mathcal{C}_1$, body codebook $\mathcal{C}_2$, hand quantizer $\mathcal{Q}^H(\cdot; \mathcal{C}^H)$, body quantizer $\mathcal{Q}^B(\cdot; \mathcal{C}^B)$ ($|\mathcal{C}^H| = |\mathcal{C}^B| = K$), H²VQ-VAE, the input motion $\mathbf{m}$, the optimization iterations $I_{max}$.

**Output:** The optimized H²VQ-VAE network $\Theta$, codebooks $\mathcal{C}^H, \mathcal{C}^B$.

**for** $I = 0, 1, \ldots, I_{\max}$ **do**

    $\mathbf{z}^H = \texttt{Enc}_H(\mathbf{m}^H)$;

    $\mathbf{z}^B = \texttt{Enc}_B(\mathbf{m}^B)$;

    $\hat{\mathbf{z}}^H = \mathcal{Q}^H(\mathbf{z}^H; \mathcal{C}^H)$;

    $\hat{\mathbf{z}}^B = \mathcal{Q}^B(\texttt{Conv1d}(\texttt{Concat}(\texttt{Transform}(\hat{\mathbf{z}}^H), \mathbf{z}^B)); \mathcal{C}^B)$;

    $\hat{\mathbf{m}} = \texttt{Dec}(\hat{\mathbf{z}}^B, \hat{\mathbf{z}}^H)$;

    $\Theta = \Theta - \nabla_\Theta \|\mathbf{m} - \texttt{Dec}(\hat{\mathbf{m}})\|_2^2 + \alpha \left(\|\mathbf{z}^H - \texttt{sg}(\hat{\mathbf{z}}^H)\|_2^2 + \|\mathbf{z}^B - \texttt{sg}(\hat{\mathbf{z}}^B)\|_2^2\right)$;

    Optimize two codebooks $\mathcal{C}^H, \mathcal{C}^B$ via $\texttt{EMA}$ and $\texttt{Code Reset}$;

**return** H²VQ-VAE network $\Theta$.

---

**Algorithm 2:** Inference procedure of Holistic Hierarchical VQ-VAE (H²VQ-VAE)

---

**Input:** The pre-trained H²VQ-VAE network $\Theta$, body and hand code indices sequence $\mathbf{I}^B = [\mathbf{I}_1^B, \mathbf{I}_2^B, \cdots, \mathbf{I}_{L/r}^B, \texttt{End}]$ and $\mathbf{I}^H = [\mathbf{I}_1^H, \mathbf{I}_2^H, \cdots, \mathbf{I}_{L/r}^H, \texttt{End}]$, codebook $\mathcal{C}^H = \{k, \mathbf{e}_1(k)\}_{k \in [K]}, \mathcal{C}^B = \{k, \mathbf{e}_2(k)\}_{k \in [K]}$.

**Output:** the noise prediction network $\epsilon_\theta$.

$\hat{\mathbf{z}}^H \leftarrow \texttt{Query } \mathcal{C}^H \text{ with } \mathbf{I}^H$;

$\hat{\mathbf{z}}^B \leftarrow \texttt{Query } \mathcal{C}^B \text{ with } \mathbf{I}^B$;

**return** motion $\texttt{Dec}(\hat{\mathbf{z}}^B, \hat{\mathbf{z}}^H)$.

---

### C.2 Comparsion with Residual Vector Quantization (RVQ)

As can be seen in Appendix C.1, our H²VQ consists of two codebooks $\mathcal{C}^H$ and $\mathcal{C}^B$ with size $K$. The intuitive design insight is that the space of our code combination is $\mathcal{O}(K^2)$. However, scaling the size of the codebook to $2K$ only has the vector space of size $\mathcal{O}(K)$. Therefore, our H²VQ enjoys the scaling of latent code size with low memory cost. An alternative way to scale the codebook size efficiently is the 2-level Residual Vector Quantization (RVQ) technique. As shown in Algorithm 3, RVQ quantized the residual error vectors recurrently in each level, which is also a hierarchical modeling strategy. However, RVQ does not model the hand and body motions explicitly, which makes it cannot reconstruct the whole-body motions better than H²VQ. For more details, please refer to Zeghidour et al. (2021). The experimental comparisons are in Appendix G.

---

**Algorithm 3:** Residual Vector Quantization (RVQ)

---

**Input:** The output of the encoder $\mathbf{z} = \texttt{Enc}(\mathbf{m})$, $N_q$-level quantizers $\mathcal{Q}_i(\cdot)$ ($i = 1, 2, \cdots, N_q$).

**Output:** Quantized vector $\hat{\mathbf{z}}$.

$\hat{\mathbf{z}} = 0$;

$\mathbf{res} = \mathbf{z}$;

**for** $i = 1, 2, \ldots, N_q$ **do**

    $\hat{\mathbf{z}} \mathrel{+}= \mathcal{Q}_i(\mathbf{res})$;

    $\mathbf{res} \mathrel{-}= \mathcal{Q}_i(\mathbf{res})$;

**return** $\hat{\mathbf{z}}$.

---

# D  TRAINING DETAILS AND EVALUATION ON TEXT-MOTION RETRIEVAL PRE-TRAINING

In this section, we will detail the training details of the TMR model and evaluate our pre-trained retrieval model.

## D.1  TRAIN DETAILS

Here, we detail the training procedure on how to train a text-whole-body-motion retrieval model. Recall a text-to-motion model, TEMOS (Petrovich et al., 2022), the VAE-based architecture consists of a motion encode, a text encoder, and a motion decoder. The training objective in TEMOS is the weighted sum of $\mathcal{L}_T = \mathcal{L}_{rec} + \lambda_{KL}\mathcal{L}_{KL} + \lambda_E\mathcal{L}_E$, where the three loss items are reconstruction loss, Kullback-Leibler (KL) divergence loss, and cross-modal embedding similarity loss respectively. Additionally, like Petrovich et al. (2023), we introduce an InfoNCE (Oord et al., 2018) loss term $\mathcal{L}_{NCE}$ into the optimization objective for learning text-motion-aligned representations. The InfoNCE loss aims to align pairwise text-motion embeddings and pull the negative motion-text pairs in the batch away like Radford et al. (2021). Therefore, the final training objective is

$$\min \mathcal{L}_T + \lambda_{NCE}\mathcal{L}_{NCE},$$

where all hyper-parameters are $\lambda_{KL} = 1 \times 10^{-5}, \lambda_E = 1 \times 10^{-5}, \lambda_{NCE} = 1 \times 10^{-1}$ respectively.

Note that, in a batch, different motion samples might be similar or even repetitive. Therefore, we will filter the similar negative samples in the InfoNCE loss. In other words, two motions with similar text descriptions (similarity higher than 0.85) will not be treated as negative samples. Technically, a pre-trained language model will calculate the similarity between two text descriptions $s_{i,j} = \langle \mathbf{t}_i, \mathbf{t}_j \rangle$, where $\langle \cdot, \cdot \rangle$ denotes the cosine similarity. Different from Petrovich et al. (2023) choosing MPNet[1] (Song et al., 2020) as the pre-trained language model, we take the Sentence-BERT (*aka* sBERT[2]) (Reimers & Gurevych, 2019) as the pre-trained language model, which is more accurate than MPNet.

To compare the accuracy of evaluating the similarity among sentences, we present a case study composed of 10 sentence samples in Example 1.

**Example 1**  *Here, we present the 10 sentence samples used for evaluating sBERT and MPNet.*

```
[
    0: 'A human walking backwards.',
    1: 'A person is walking backwards.',
    2: 'Someone walks in a circle counterclockwise',
    3: 'A person walks a full counter-clockwise circle.',
    4: 'A human performs a tight 90° curve to the right.',
    5: 'A person walks a quarter circle clockwise with 4 steps.',
    6: 'human goes backwards starting with left',
    7: 'A person walks backwards.',
    8: 'a person walks in a circle to the left side.',
    9: 'trump'
]
```

We calculate the cosine similarity of these 10 sentences with sBERT and MPNet, respectively. As shown in Figure 6, sBERT reflects the sentence similarity more accurately than MPNet. For two sentences with very similar semantics, like 'A human walking backwards.' and 'A person is walking backwards.', the similarity provided by sBERT is 0.958, while MPNet is 0.893. For two sentences completely unrelated, like 'A human walking backwards.' and 'trump', the similarity provided by sBERT is 0.132, while MPNet is 0.758. In this case, the 'trump' example is not a motion description. sBERT clearly distinguishes it from other sentences, but MPNet cannot distinguish them significantly. Therefore, the sBERT is more discriminative than MPNet in negative filtering.

---

[1]https://huggingface.co/microsoft/mpnet-base.
[2]https://huggingface.co/sentence-transformers/all-MiniLM-L6-v2.

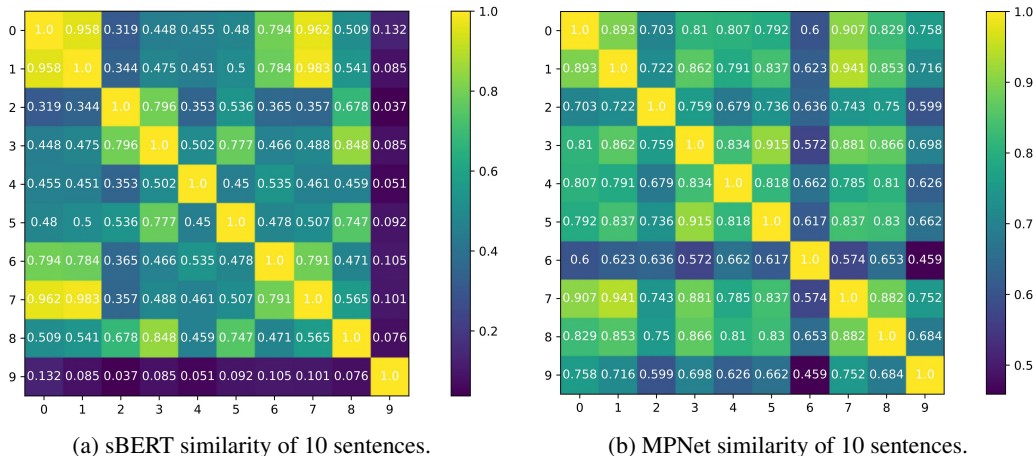

(a) sBERT similarity of 10 sentences.  (b) MPNet similarity of 10 sentences.

Figure 6: Sentences similarity comparison between the sBERT and MPNet .

## D.2 EVALUATION OF THE RETRIEVAL MODEL

We take the Recall@$K$ as the main metric to evaluate the retrieval performance to evaluate the performance of the TMR model. We evaluate both motion-to-text (M2T) and text-to-motion (T2M) retrieval performance with four main protocols. (A) *Retrieving in the full test test.* (B) *Retrieving in the full test test with a sBERT-score threshold (set $\epsilon$ as 0.9).* As some sentences have similar semantics, like "A man is walking straight." and "The person walks forward.", we treat these retrieval results as positive targets if the retrieved text has a sBERT similarity higher than $\epsilon = 0.9$ with GT text. (C) *Retrieving in the 256-size sub-test set.* The 256-size retrieving set consists of one GT result and 255 negative results. (D) *Retrieving in the 32-size sub-test set.* Similar to Protocol C, the 32-size retrieving set consists of one GT result and 31 negative results. The T2M and M2T retrieval evaluation results on Motion-X are shown in Table 5. The T2M and M2T retrieval evaluation results on HumanML3D are shown in Table 6. The comparison with other text-motion retrieval methods on the protocol (C) and protocol (D) is shown in Appendix I. **The retrieval web demo on both body-only and whole-body datasets will be public if accepted.**

|  | T2M | | | | | M2T | | | | |
|---|---|---|---|---|---|---|---|---|---|---|
|  | Recall@1 | Recall@2 | Recall@3 | Recall@5 | Recall@10 | Recall@1 | Recall@2 | Recall@3 | Recall@5 | Recall@10 |
| Protocol A | 0.051 | 0.098 | 0.131 | 0.192 | 0.301 | 0.066 | 0.118 | 0.163 | 0.233 | 0.350 |
| Protocol B | 0.089 | 0.152 | 0.194 | 0.273 | 0.401 | 0.169 | 0.205 | 0.238 | 0.298 | 0.395 |
| Protocol C | 0.445 | 0.609 | 0.700 | 0.799 | 0.883 | 0.407 | 0.578 | 0.673 | 0.795 | 0.883 |
| Protocol D | 0.716 | 0.854 | 0.907 | 0.946 | 0.977 | 0.771 | 0.893 | 0.938 | 0.968 | 0.985 |

Table 5: Recall@$K$ (T2M and M2T) of GT motions and texts on the Motion-X dataset.

|  | T2M | | | | | M2T | | | | |
|---|---|---|---|---|---|---|---|---|---|---|
|  | Recall@1 | Recall@2 | Recall@3 | Recall@5 | Recall@10 | Recall@1 | Recall@2 | Recall@3 | Recall@5 | Recall@10 |
| Protocol A | 0.065 | 0.117 | 0.155 | 0.227 | 0.339 | 0.057 | 0.106 | 0.144 | 0.205 | 0.322 |
| Protocol B | 0.204 | 0.282 | 0.326 | 0.404 | 0.510 | 0.102 | 0.151 | 0.199 | 0.263 | 0.373 |
| Protocol C | 0.359 | 0.523 | 0.630 | 0.729 | 0.842 | 0.365 | 0.527 | 0.625 | 0.731 | 0.838 |
| Protocol D | 0.774 | 0.896 | 0.937 | 0.968 | 0.985 | 0.711 | 0.853 | 0.905 | 0.947 | 0.977 |

Table 6: Recall@$K$ (T2M and M2T) of GT motions and texts on the HumanML3D dataset.

### D.3 RETRIEVAL ABILITY COMPARISON (TMR *v.s.* TEMOS)

To verify the good alignment of TMR, we compare its retrieval ability with TEMOS Petrovich et al. (2022). As shown in Table 7, the TMR enjoys a good alignment between texts and motions by the contrastive training objective, which makes it with a larger margin than TEMOS in retrieval. This good retrieval ability provides a better alignment of two modalities, and provide a better motion-text alignment for motion generation.

| Protocol | Model | T2M | | | | | M2T | | | | |
|---|---|---|---|---|---|---|---|---|---|---|---|
| | | Recall@1 | Recall@2 | Recall@3 | Recall@5 | Recall@10 | Recall@1 | Recall@2 | Recall@3 | Recall@5 | Recall@10 |
| A | TEMOS | 0.034 | 0.062 | 0.082 | 0.117 | 0.178 | 0.034 | 0.067 | 0.096 | 0.137 | 0.204 |
| A | TMR | **0.051** | **0.098** | **0.131** | **0.192** | **0.301** | **0.066** | **0.118** | **0.163** | **0.233** | **0.350** |
| B | TEMOS | 0.115 | 0.163 | 0.190 | 0.237 | 0.308 | 0.112 | 0.135 | 0.155 | 0.190 | 0.246 |
| B | TMR | **0.089** | **0.152** | **0.194** | **0.273** | **0.401** | **0.169** | **0.205** | **0.238** | **0.298** | **0.395** |
| C | TEMOS | 0.233 | 0.341 | 0.412 | 0.492 | 0.601 | 0.263 | 0.360 | 0.426 | 0.504 | 0.614 |
| C | TMR | **0.445** | **0.609** | **0.700** | **0.799** | **0.883** | **0.407** | **0.578** | **0.673** | **0.795** | **0.883** |
| D | TEMOS | 0.502 | 0.641 | 0.717 | 0.802 | 0.897 | 0.528 | 0.654 | 0.725 | 0.807 | 0.907 |
| D | TMR | **0.716** | **0.854** | **0.907** | **0.946** | **0.977** | **0.771** | **0.893** | **0.938** | **0.968** | **0.985** |

Table 7: The Recall@$K$ (T2M and M2T) of GT motions and texts on the Motion-X dataset (TMR *v.s.* TEMOS).

# E FAILURE CASES OF BASELINES

As discussed in Section 2.4, previous methods shown in Figure 3 will fail in some scenarios. We discuss the fashion of "Supervision by an image-text aligned prior explicitly" (Figure 3a) here. As there was no strong text-motion-aligned pre-trained model, MotionCLIP (Tevet et al., 2022) renders the generated motions as images and then supervises the alignment between text and rendered images with the CLIP model. This supervision will cause the generated motion to be *over-smoothing*, even *stillness*. We show some over-smoothing cases[3,4] of MotionCLIP on here. As shown in Figure 7, there is almost no change at all between the first frame (Figure 7a) and the final frame (Figure 7b) of motion.

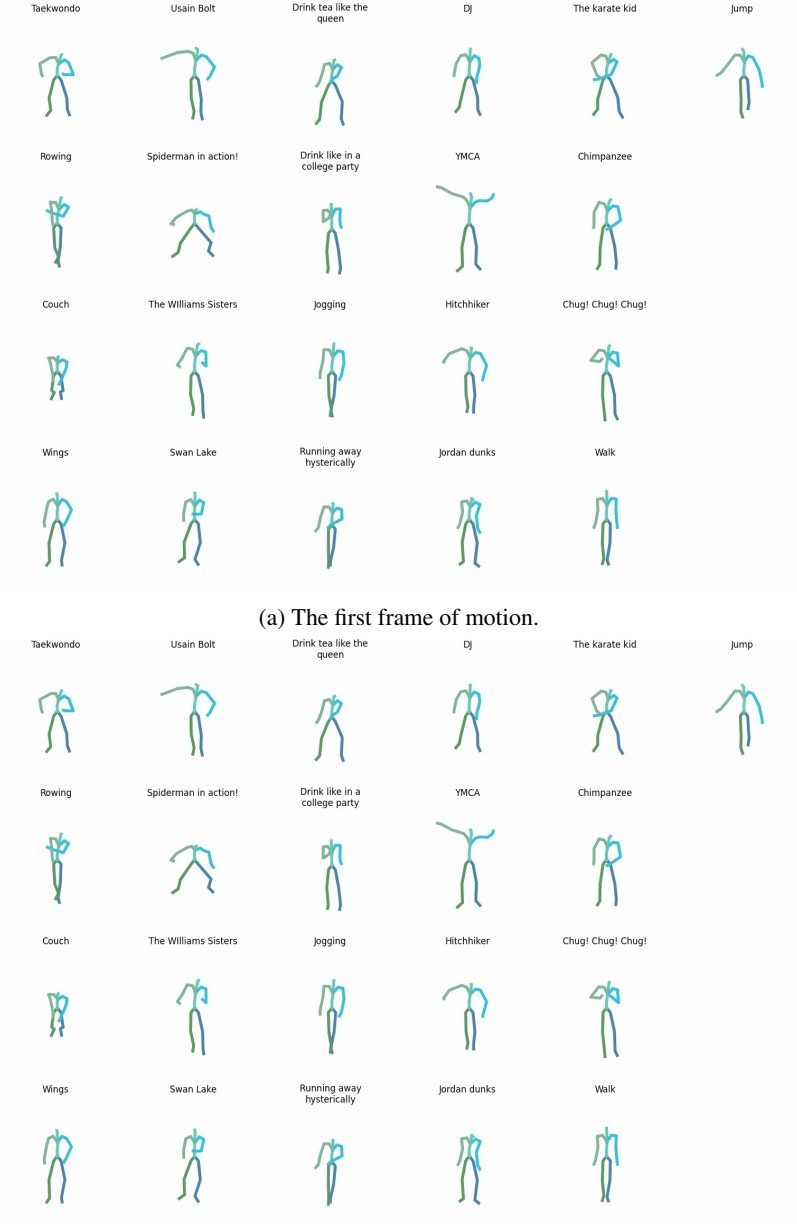

(a) The first frame of motion.

(b) The final frame of motion.

Figure 7: Visualization of MotionCLIP generated results. The first frame and the final frame of motions are shown in the figure.

---

[3]https://github.com/GuyTevet/MotionCLIP/issues/5.
[4]https://github.com/GuyTevet/MotionCLIP/issues/15.

## F  MORE DETAILS ON MAIN RESULTS (RQ1)

### F.1  QUANTITATIVE COMPARISON

In the main paper, we report the metrics of our HumanTOMATO and related baselines in Table 1. We repeat the evaluation 5 times and report the `mean`$^{\pm \mathtt{std}}$ results in Table 8 and Table 9. Experimental results show our strength than baseline on generation quality, text-motion alignment, and diversity.

| | FID↓ | R-Precision$^{(32)}$ | | | TMR-R-Precision$^{(256)}$ | | |
| --- | --- | --- | --- | --- | --- | --- | --- |
| | | Top1↑ | Top2↑ | Top3↑ | Top1↑ | Top2↑ | Top3↑ |
| GT | - | $0.500^{\pm 0.002}$ | $0.708^{\pm 0.002}$ | $0.814^{\pm 0.002}$ | $0.407^{\pm 0.003}$ | $0.578^{\pm 0.004}$ | $0.673^{\pm 0.003}$ |
| T2M-GPT | $1.366^{\pm 0.059}$ | $0.368^{\pm 0.005}$ | $0.553^{\pm 0.003}$ | $0.655^{\pm 0.007}$ | $0.310^{\pm 0.001}$ | $0.446^{\pm 0.007}$ | $0.527^{\pm 0.014}$ |
| MDM | $3.800^{\pm 0.020}$ | $0.352^{\pm 0.003}$ | $0.547^{\pm 0.002}$ | $0.634^{\pm 0.004}$ | $0.310^{\pm 0.004}$ | $0.430^{\pm 0.007}$ | $0.530^{\pm 0.014}$ |
| MLD | $3.407^{\pm 0.020}$ | $0.385^{\pm 0.002}$ | $0.571^{\pm 0.001}$ | $0.683^{\pm 0.001}$ | $0.333^{\pm 0.004}$ | $0.477^{\pm 0.003}$ | $0.561^{\pm 0.001}$ |
| HumanTOMATO | $\mathbf{1.174}^{\pm 0.015}$ | $\mathbf{0.416}^{\pm 0.009}$ | $\mathbf{0.603}^{\pm 0.007}$ | $\mathbf{0.703}^{\pm 0.007}$ | $\mathbf{0.399}^{\pm 0.000}$ | $\mathbf{0.555}^{\pm 0.005}$ | $\mathbf{0.638}^{\pm 0.004}$ |

Table 8: Quantitative Comparison on the Motion-X dataset (FID, TMR-R-Precision$^{(256)}$, and R-Precision$^{(32)}$ metrics).

| | TMR-Matching Score↓ | Matching Score↓ | MModality↑ | Diversity↑ |
| --- | --- | --- | --- | --- |
| GT | $0.768^{\pm 0.000}$ | $2.888^{\pm 0.006}$ | - | $11.087^{\pm 0.271}$ |
| T2M-GPT | $0.881^{\pm 0.004}$ | $4.316^{\pm 0.053}$ | $2.356^{\pm 0.093}$ | $10.753^{\pm 0.063}$ |
| MDM | $0.840^{\pm 0.004}$ | $4.050^{\pm 0.023}$ | $\mathbf{2.530}^{\pm 0.041}$ | $\mathbf{11.400}^{\pm 0.370}$ |
| MLD | $0.883^{\pm 0.002}$ | $3.901^{\pm 0.011}$ | $2.448^{\pm 0.034}$ | $10.420^{\pm 0.234}$ |
| HumanTOMATO | $\mathbf{0.809}^{\pm 0.002}$ | $\mathbf{3.894}^{\pm 0.008}$ | $1.732^{\pm 0.194}$ | $10.812^{\pm 0.034}$ |

Table 9: Quantitative Comparison on the Motion-X dataset (TMR-Matching Score, Matching Score, and MModality metrics).

We provide the facial motion generation metrics in Table 10. Our facial motions have expressive capabilities and satisfy the textual descriptions. Nonetheless, the quality of generated facial motions and alignment with text could be improved. We leave this as our future work.

| | FID | Top1 | Top2 | Top3 | Matching-score | MModality | Diversity |
| --- | --- | --- | --- | --- | --- | --- | --- |
| GT | - | $0.351^{\pm 0.001}$ | $0.605^{\pm 0.001}$ | $0.776^{\pm 0.001}$ | $1.159^{\pm 0.00\dot{}}$ | - | $11.661^{\pm 0.091}$ |
| HumanTOMATO | $21.418^{\pm 0.038}$ | $0.162^{\pm 0.001}$ | $0.283^{\pm 0.001}$ | $0.372^{\pm 0.001}$ | $6.643^{\pm 0.003}$ | $0.570^{\pm 0.025}$ | $8.098^{\pm 0.085}$ |

Table 10: Facial motion generation results on Motion-X dataset.

In the main paper, we compare our method with baseline methods with key-frame sequence visualization. We provide more comparison in Figure 8. In Figure F, the lighter colors represent earlier snapshots. As can be seen, T2M-GPT lacks temporal sensitivity and will generate motions that do not match the text description. In contrast, our method will enjoy these scenarios well and generate vivid motions well aligned with texts.

Additionally, we visualize more generated results of HumanTOMATO in Figure 9 and Figure 10, which show our good generation performance.

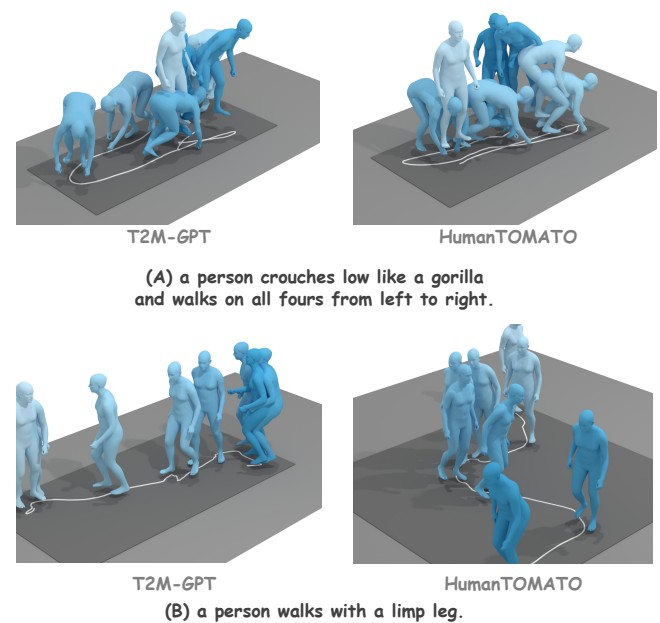

(A) a person crouches low like a gorilla
and walks on all fours from left to right.

(B) a person walks with a limp leg.

Figure 8: Qualitative comparison with T2M-GPT.

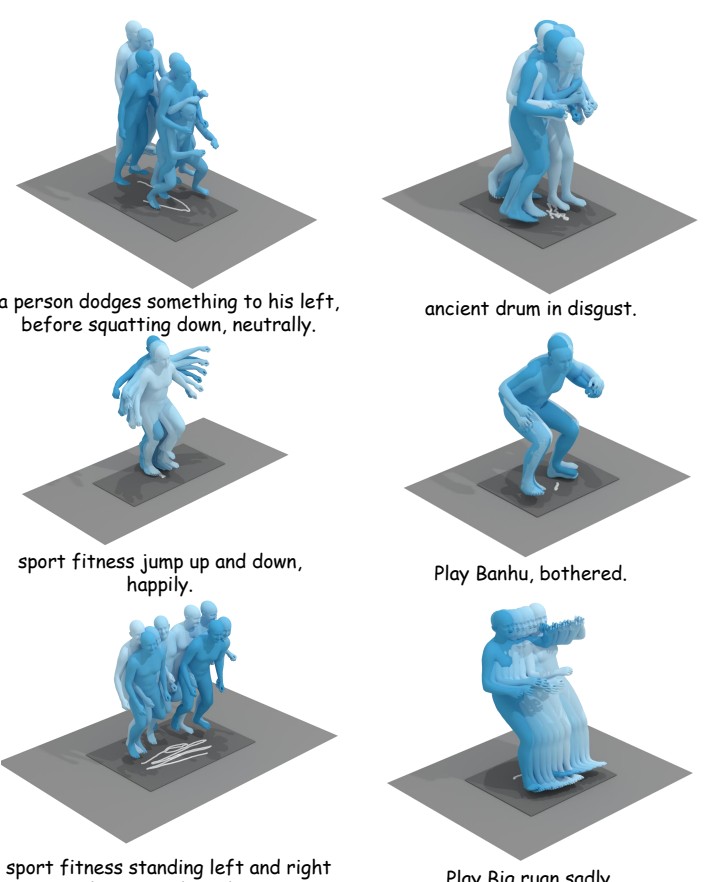

a person dodges something to his left,
before squatting down, neutrally.

ancient drum in disgust.

sport fitness jump up and down,
happily.

Play Banhu, bothered.

sport fitness standing left and right
leg swing, happily.

Play Big ruan, sadly.

Figure 9: Visualization of the whole-body motions generated by HumanTOMATO (1).

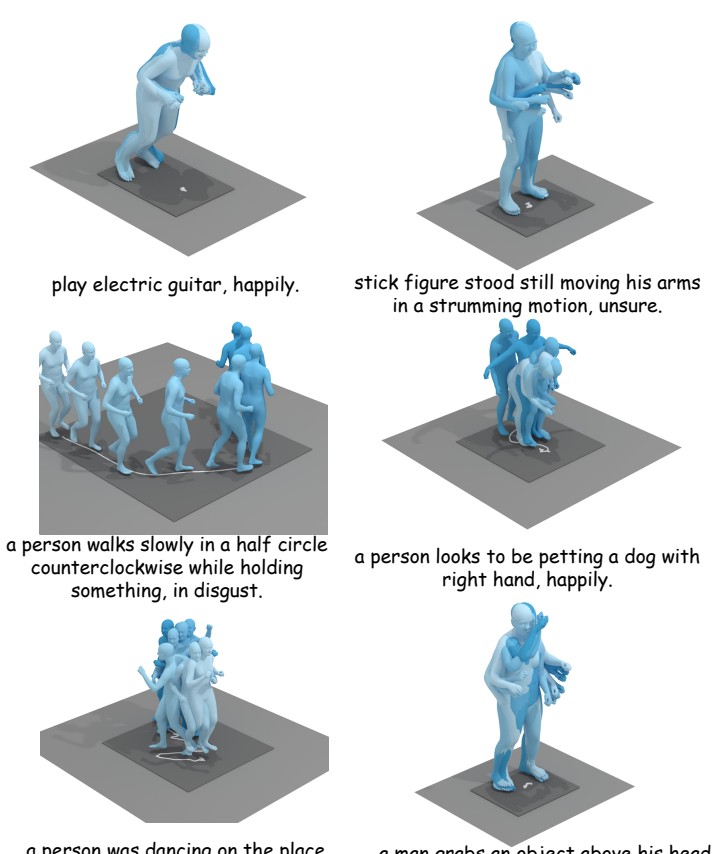

play electric guitar, happily.

stick figure stood still moving his arms in a strumming motion, unsure.

a person walks slowly in a half circle counterclockwise while holding something, in disgust.

a person looks to be petting a dog with right hand, happily.

a person was dancing on the place while rasing the hands up, sadly.

a man grabs an object above his head with his right hand, sadly.

Figure 10: Visualization of the whole-body motions generated by HumanTOMATO (2).

## G COMPARISON ON DIFFERENT VECTOR QUANTIZATION METHODS (RQ2)

### G.1 ABLATION ON DIFFERENT QUANTIZATION METHODS

In the main paper (Section 3.4), we report the MPJPE for evaluating the reconstruction error of Vanilla VQ, RVQ, and $H^2$VQ respectively. Although HumanML3D only includes body-part motions, we compare the Vinilla VQ-VAE with the RVQ technique to verify our motivation on hierarchical motion modeling, whose results are shown in Table 13. Additionally, as shown in Table 11 and Table 12, we provide more evaluation metrics on PA-MPJPE and Acceleration error (Accel.) (Gower, 1975; Lin et al., 2023a; Zeng et al., 2022; Chen et al., 2023b). to evaluate the reconstruction quality. Evaluation results show that naïvely increasing the codebook size is almost in vain, and hierarchical modeling is effective for action modeling. Besides, our $H^2$VQ is a better design on whole-body motions than RVQ.

| | MPJPE | | | PA-MPJPE | | | Accel. | | |
|---|---|---|---|---|---|---|---|---|---|
| | All↓ | Body↓ | Hand↓ | All↓ | Body↓ | Hand↓ | All↓ | Body↓ | Hand↓ |
| Vanilla VQ (512) | 140.66 | 92.20 | 46.45 | 58.23 | 47.72 | 17.03 | 23.73 | 19.99 | 26.46 |
| Vanilla VQ (1024) | 139.33 | 91.77 | 46.40 | 57.30 | 46.79 | 17.01 | 23.54 | 19.71 | 26.35 |
| RVQ | 110.94 | 73.97 | 40.01 | 40.63 | 35.84 | 14.46 | 21.22 | 17.76 | 23.75 |
| $H^2$VQ | **92.97** | **62.34** | **37.20** | **34.21** | **30.76** | **14.05** | **18.95** | **16.53** | **20.72** |

Table 11: Different vector quantization methods on Motion-X.

| | MPJPE | | | PA-MPJPE | | | Accel. | | |
|---|---|---|---|---|---|---|---|---|---|
| | All↓ | Body↓ | Hand↓ | All↓ | Body↓ | Hand↓ | All↓ | Body↓ | Hand↓ |
| Vanilla VQ (512) | 78.23 | 38.29 | 31.48 | 35.32 | 21.75 | 14.51 | 11.01 | 7.32 | 13.71 |
| Vanilla VQ (1024) | 76.01 | 37.34 | 29.89 | 33.42 | 20.92 | 14.14 | 10.70 | 7.23 | 13.25 |
| RVQ | 62.94 | 31.12 | 27.28 | 25.61 | 15.96 | **13.06** | **8.80** | 6.67 | **10.37** |
| $H^2$VQ | **46.74** | **24.33** | **24.59** | **22.00** | **13.95** | 13.48 | 10.11 | **6.05** | 13.09 |

Table 12: Different vector quantization methods on GRAB.

| | MPJPE (Body)↓ | PA-MPJPE (Body)↓ | Accel. (Body)↓ |
|---|---|---|---|
| Vanilla VQ (512) | 77.209 | 45.53 | 8.36 |
| Vanilla VQ (1024) | 71.34 | 40.75 | 7.59 |
| RVQ | **63.05** | **30.99** | **6.46** |

Table 13: Different vector quantization methods on HumanML3D.

We additionally discuss how the $H^2$VQ helps the motion generation from the aspect of motion quality and text-motion alignment. We take the T2M-GPT as the baseline and compare it to the hierarchical reconstruction setting. The difference between the two settings is with or without the $H^2$VQ method. As shown in Table 14, the $H^2$VQ helps both motion generation and text-motion alignment significantly.

| | FID↓ | R-Precision[(32)] | | | TMR-R-Precision[(256)] | | | TMR-Matching Score ↓ | Matching Score ↓ |
|---|---|---|---|---|---|---|---|---|---|
| | | Top1↑ | Top2↑ | Top3↑ | Top1↑ | Top2↑ | Top3↑ | | |
| GT | - | 0.500 | 0.708 | 0.814 | 0.407 | 0.578 | 0.673 | 0.768 | 2.888 |
| T2M-GPT w/o $H^2$VQ | 1.366 | 0.368 | 0.553 | 0.655 | 0.310 | 0.446 | 0.527 | 0.881 | 4.316 |
| T2M-GPT w/ $H^2$VQ | **1.086** | **0.405** | **0.588** | **0.695** | **0.345** | **0.490** | **0.573** | **0.844** | **3.917** |

Table 14: The ablation on how can $H^2$VQ help the whole-body motion generation on T2M-GPT.

We show more visualization results here. Our method excels in two perspectives, body-part reconstruction and hand-part reconstruction. On the one hand, From 11a, our method $H^2$VQ in the middle column achieves a significantly higher level of accuracy in reconstructing global translation. From 11b, our method could perform better on movement direction reconstruction and motion coherence. From 11c, our method could reconstruct motion more precisely than other methods even with minor motion movements. On the other hand, because of our decoupled design, our method performs better on hand movement and pose reconstruction. As shown in 12, ours (in blue) can precisely reconstruct the GT hand pose (in green), while the Vanilla VQ-VAE method fails in most of these cases, which demonstrates the superiority of our design.

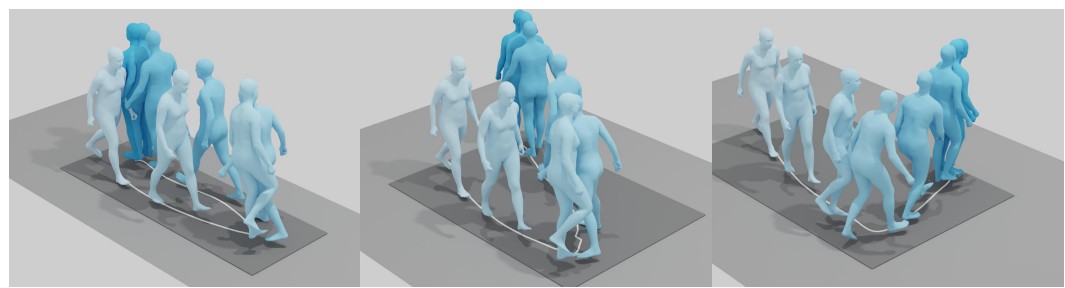

(a) Case 1. H$^2$VQ performs better on trajectory reconstruction. (GT, H$^2$VQ, and Vanilla VQ)

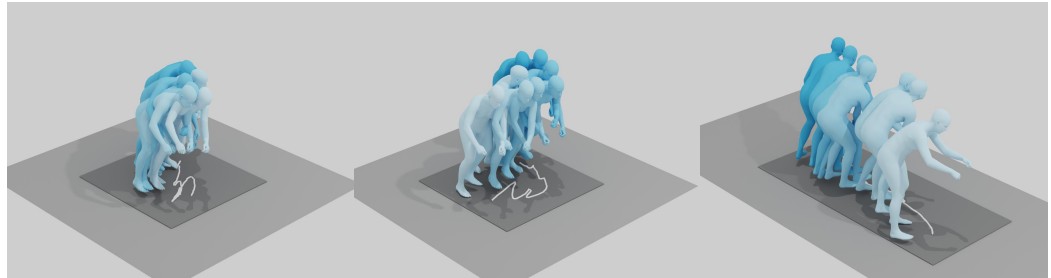

(b) Case 2. H$^2$VQ performs better on direction reconstruction and motion coherence. (GT, H$^2$VQ, and Vanilla VQ)

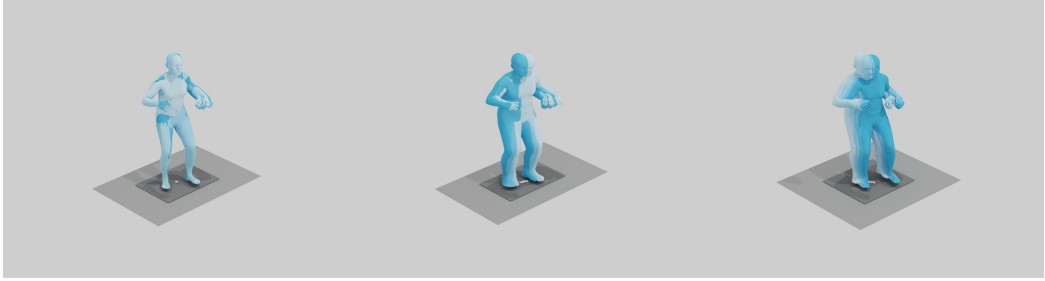

(c) Case 3. H$^2$VQ performs better on reconstructing motions with low amplitude. (GT, H$^2$VQ, and Vanilla VQ)

Figure 11: Visualization of motion reconstruction on the Motion-X dataset (body motion reconstruction perspective). From the left to right are GT, H$^2$VQ, and Vanilla VQ, respectively.

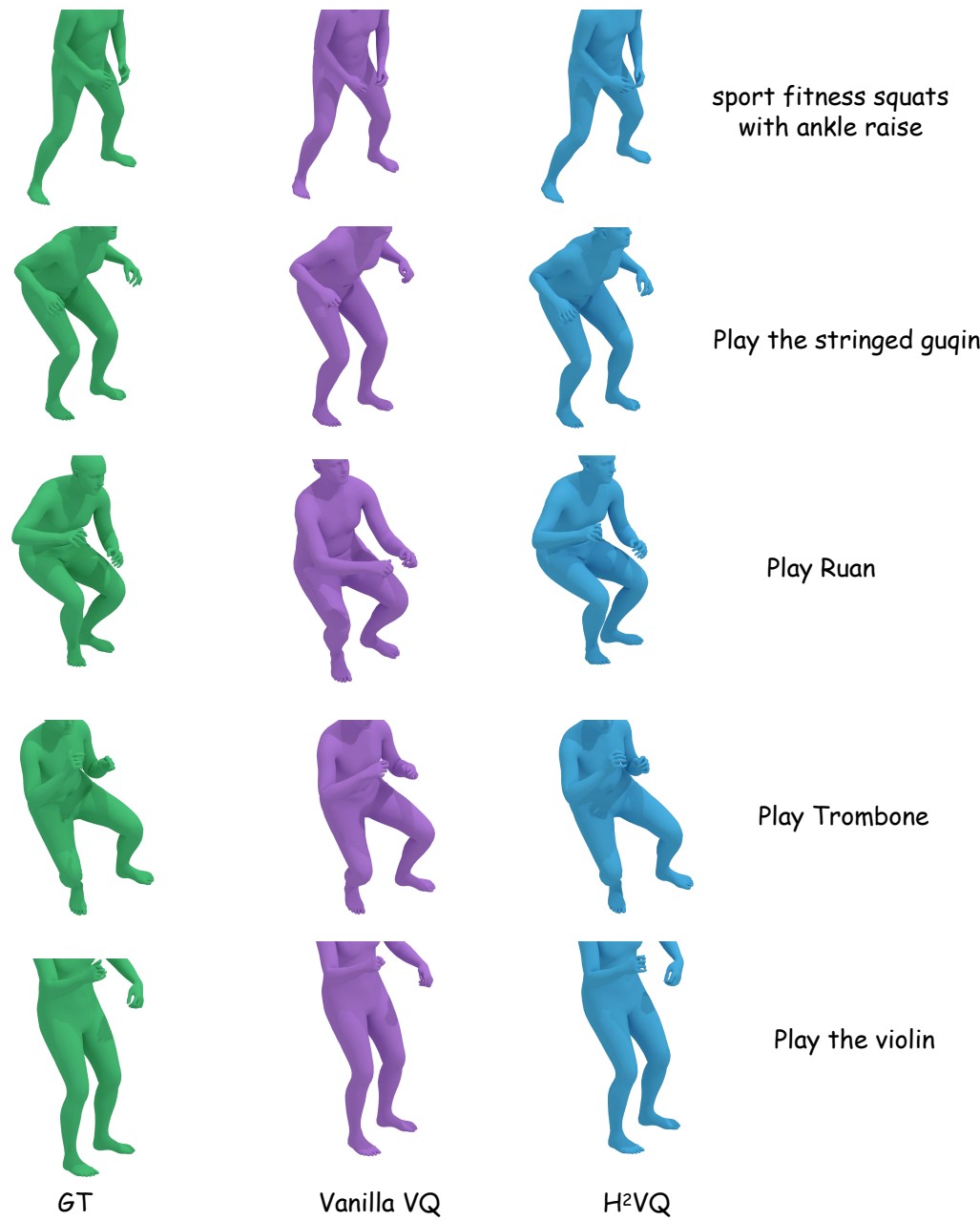

sport fitness squats
with ankle raise

Play the stringed guqin

Play Ruan

Play Trombone

Play the violin

GT          Vanilla VQ          H²VQ

Figure 12: Visualization of motion reconstruction on the Motion-X dataset (hands motion reconstruction perspective). From the left to right are GT, Vanilla VQ, and H$^2$VQ, respectively.

### G.2 Comparisons on different Codebook Sizes

We discuss how much the scaling of codebook size benefits the generation results. We perform the comparison on Vanilla VQ, RVQ, and $H^2$VQ. As shown in Figure 13, $H^2$VQ performs best among the three quantization methods. When doubling the codebook size, the final reconstruction error (MPJPE) reduces marginally. This verifies that scaling of codebook size in VQ-VAE is almost in vain. This observation supports the basic intuition on the designing of $H^2$VQ.

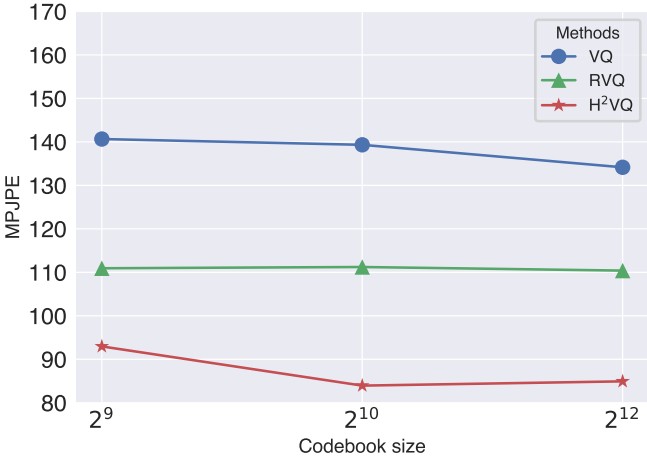

Figure 13: The ablation on the codebook size. Reconstruction results of GT, Vanilla VQ, and $H^2$VQ are presented respectively.

# H  TEXT-MOTION RETRIEVAL MODEL AS A PRIOR (RQ3)

## H.1  QUANTITATIVE RESULTS ON HUMANML3D

In the main paper, we verify that the pre-trained text-motion-aligned model provides a strong prior to text-aligned whole-body motion generation. Additionally, the text-motion-aligned prior not only benefits the whole-body motion generation but also helps the text-motion alignment in body-only motion generation. We take the T2M-GPT as baseline (line 1 in the Table 15), and we ablate whether the TMR language embedding and text-motion-alignment supervision help to generate the text-aligned body-only motions. As shown in Table 15, our experiments on HumanML3D show that both the motion-aware language prior and the text-motion-alignment supervision help to generate higher quality and text-aligned motions (on FID and TMR-R-Precision$^{(256)}$).

| embedding | supervision | FID ↓ | TMR-R-Precision$^{(256)}$ | | | R-Precision$^{(256)}$ | | | TMR-Matching-score ↓ | Matching-score ↓ |
|---|---|---|---|---|---|---|---|---|---|---|
| | | | Top1 ↑ | Top2 ↑ | Top3 ↑ | Top1 ↑ | Top2 ↑ | Top3 ↑ | | |
| CLIP | ✗ | 0.474 | 0.082 | 0.129 | 0.168 | 0.169 | 0.259 | 0.341 | 1.322 | 3.155 |
| TMR | ✗ | 0.326 | 0.147 | 0.206 | 0.269 | 0.177 | 0.281 | 0.396 | 1.285 | 2.915 |
| TMR | ✔ | **0.312** | **0.159** | **0.223** | **0.276** | **0.184** | **0.292** | **0.396** | **1.282** | **2.906** |

Table 15: Abaltion on how pre-trained text-motion aligned model helps to generate the text-aligned body-only motion (on HumanML3D).

## H.2  PRE-TRAINED TEXT-MOTION RETRIEVAL MODEL AS A PRIOR

We test on the Motion-X dataset first to explore whether our text-motion-aligned text encoder helps the generated motions align well with the given text. As shown in Figure 14(a), the model with our design performs the "kick" motion. As shown in Figure 14(b) and Figure 14(c), HumanTOMATO learning with motion-aware language prior has a better understanding of motion trajectory and temporal relations.

We test some cases in the wild to explore whether our text-motion-aligned text encoder helps the generated motions align well with the given text. We show some cases for comparison in Figure 15. In Figure 15a, if T2M-GPT learns without motion-aware language prior, the person walks in a quarter of counter-clockwise circle. The model with motion-aware language prior will generate the motion well aligned with the given text on direction and trajectory. For the second case in Figure 15b, our design helps the model to generate motions much better in the motion direction. For the third case, our method is better aligned with text on the caption "back" and does not switch the left or right backward direction.

In summary, as claimed in Section 2.4, our method can understand the motion dynamic clues better on sequentiality, directions, and dynamics.

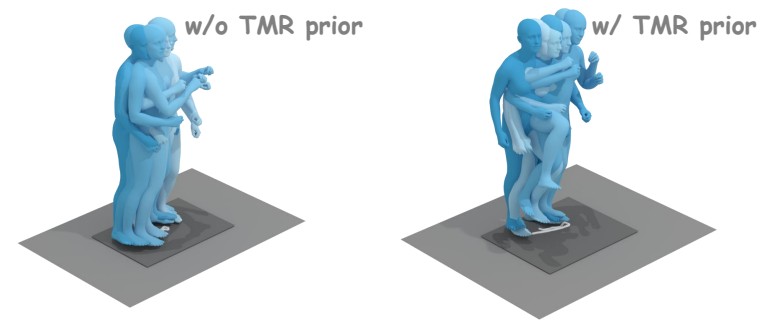

**(A) a person performs a standing kick.**

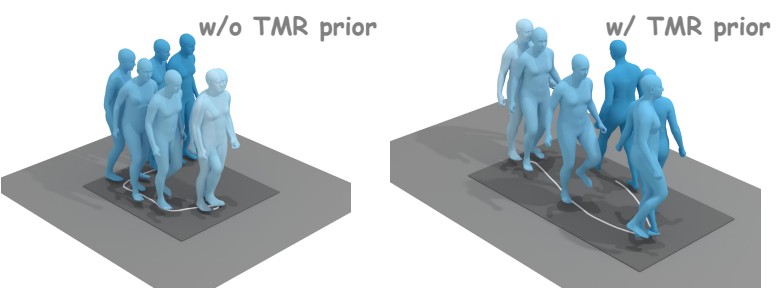

**(B) The man walks forward a couple steps, turns right 180 degrees and then walks back.**

Figure 14: Visualization on our HumanTOMATO, learning without (left) or with (right) motion-aware language prior. The left is the generated motion of HumanTOMATO without language prior, and the right is HumanTOMATO.

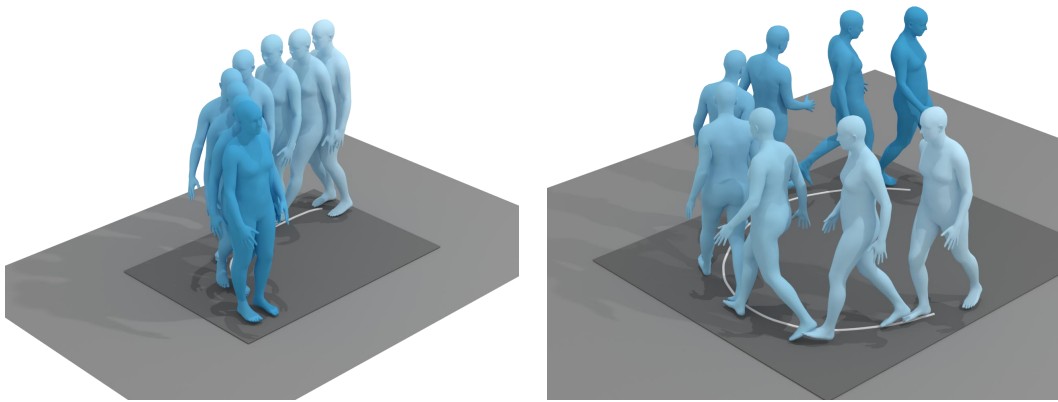

(a) Input text: "a person walks clockwisely.". The left is the generated motion of T2M-GPT, and the right is T2M-GPT learning with motion-aware language prior.

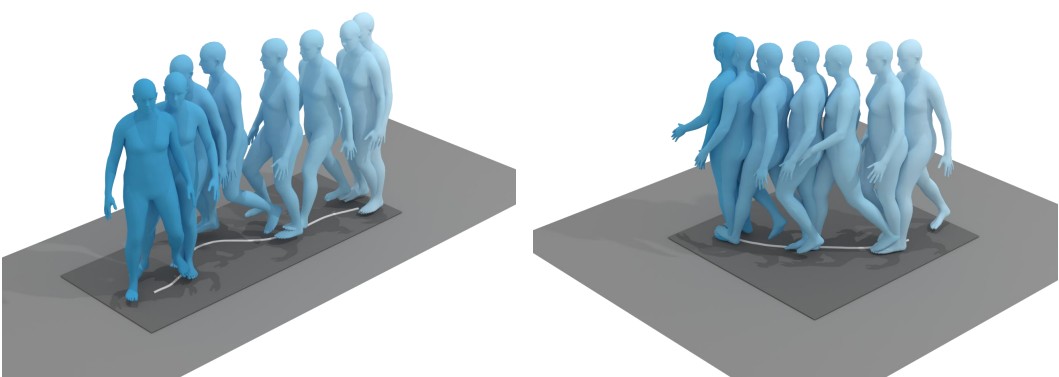

(b) Input text: "a person walks forward, turn right, finally turn right.". The left is the generated motion of T2M-GPT, and the right is T2M-GPT learning with motion-aware language prior.

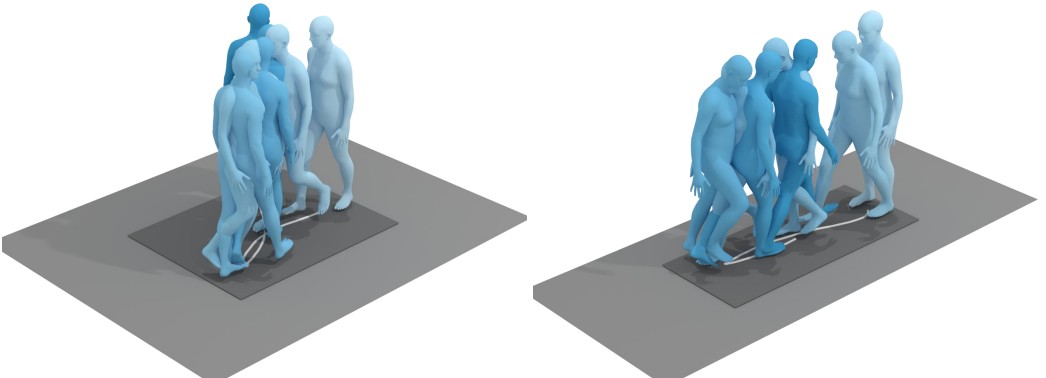

(c) Input text: "A person walks forward and then turns back.". The left is the generated motion of T2M-GPT, and the right is T2M-GPT learning with language prior.

Figure 15: Visualization on T2M-GPT, learning without (left) or with (right) motion-aware language prior. The left is the generated motion of T2M-GPT, and the right is T2M-GPT learning with language prior.

# I Details on the Evaluation Metrics (RQ4)

In Section 3.6, we analyze why the proposed evaluation metrics of alignment between generated motions and given texts are more accurate and challenging on the Motion-X dataset. Here, we provide more comparisons on both body-only and whole-body datasets to verify the universality of the proposed metrics, all of which are calculated 3 times to calculate the mean and standard value ($mean^{\pm std}$). The comparison is shown in Table 16 and Table 17. We also visualize the comparison on the HumanML3D dataset in Figure 16. Similar to the conclusion in Section 3.6, our metrics are more accurate and challenging than Guo et al. (2022a)'s in the following two aspects. (1) *TMR-R-Precision*$^{(B)}$ and *TMR-Matching-score*$^{(B)}$ metrics are more accurate than Guo et al. (2022a)'s *R-Precision*$^{(B)}$ and *Matching-score* metrics. (2) $B = 256$ is a more challenging retrieval setting than the $B = 32$ setting.

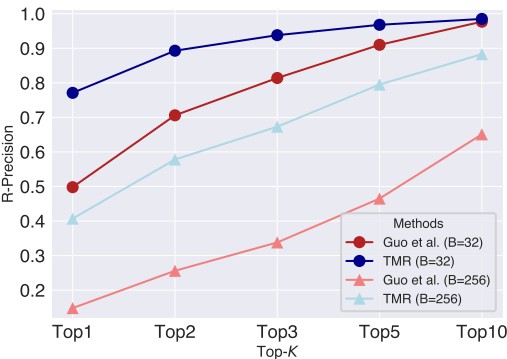

Figure 16: Comparison with existing metrics on HumanML3D. Existing evaluation metrics Guo et al. (2022a) are illustrated in red and ours are in blue. The $B = 32$ and $B = 256$ settings for retrieval are denoted as "─●─" and "─▲─" respectively.

| | Top1 | Top2 | Top3 | Top5 | Top10 |
|---|---|---|---|---|---|
| Guo et al. (2022a) $B = 32$ | $0.498^{\pm.006}$ | $0.706^{\pm.005}$ | $0.814^{\pm.003}$ | $0.910^{\pm.003}$ | $0.977^{\pm.001}$ |
| TMR $B = 32$ | $\mathbf{0.771}^{\pm.001}$ | $\mathbf{0.893}^{\pm.003}$ | $\mathbf{0.938}^{\pm.002}$ | $\mathbf{0.968}^{\pm.001}$ | $\mathbf{0.985}^{\pm.000}$ |
| Guo et al. (2022a) $B = 32$ | $0.148^{\pm.002}$ | $0.256^{\pm.004}$ | $0.338^{\pm.004}$ | $0.465^{\pm.003}$ | $0.651^{\pm.002}$ |
| TMR $B = 256$ | $\mathbf{0.407}^{\pm.003}$ | $\mathbf{0.578}^{\pm.004}$ | $\mathbf{0.673}^{\pm.003}$ | $\mathbf{0.795}^{\pm.001}$ | $\mathbf{0.883}^{\pm.001}$ |

Table 16: R-Precision of GT motions and texts on the Motion-X dataset.

| | Top1 | Top2 | Top3 | Top5 | Top10 |
|---|---|---|---|---|---|
| Guo et al. (2022a) $B = 32$ | $0.511^{\pm.003}$ | $0.705^{\pm.002}$ | $0.795^{\pm.003}$ | $0.887^{\pm.003}$ | $0.964^{\pm.003}$ |
| TMR $B = 32$ | $\mathbf{0.711}^{\pm.005}$ | $\mathbf{0.853}^{\pm.001}$ | $\mathbf{0.905}^{\pm.002}$ | $\mathbf{0.947}^{\pm.001}$ | $\mathbf{0.977}^{\pm.001}$ |
| Guo et al. (2022a) $B = 256$ | $0.167^{\pm.002}$ | $0.279^{\pm.002}$ | $0.368^{\pm.003}$ | $0.490^{\pm.004}$ | $0.659^{\pm.003}$ |
| TMR $B = 256$ | $\mathbf{0.365}^{\pm.003}$ | $\mathbf{0.527}^{\pm.002}$ | $\mathbf{0.625}^{\pm.004}$ | $\mathbf{0.731}^{\pm.003}$ | $\mathbf{0.838}^{\pm.002}$ |

Table 17: R-Precision of GT motions and texts on the HumanML3D dataset.

## J CAN WE GENERATE WHOLE-BODY MOTIONS BY PARTS SEPARATELY?

In this section, we will discuss whether we can generate whole-body motions by parts separately. To answer this question, we provide an ablation on whether to model them separately in Table 18. In Table 18, the "Modeling Separately" means modeling the hand and body motion separately.

| | FID↓ | R-Precision$^{(32)}$ | | | TMR-R-Precision$^{(256)}$ | | |
|---|---|---|---|---|---|---|---|
| | | Top1↑ | Top2↑ | Top3↑ | Top1↑ | Top2↑ | Top3↑ |
| GT | - | $0.500^{\pm 0.002}$ | $0.708^{\pm 0.002}$ | $0.814^{\pm 0.002}$ | $0.407^{\pm 0.003}$ | $0.578^{\pm 0.004}$ | $0.673^{\pm 0.003}$ |
| Modeling Separately | $2.209^{\pm 0.047}$ | $0.359^{\pm 0.002}$ | $0.551^{\pm 0.003}$ | $0.666^{\pm 0.002}$ | $0.306^{\pm 0.003}$ | $0.459^{\pm 0.002}$ | $0.552^{\pm 0.002}$ |
| HumanTOMATO | $\mathbf{1.174}^{\pm 0.015}$ | $\mathbf{0.416}^{\pm 0.009}$ | $\mathbf{0.603}^{\pm 0.007}$ | $\mathbf{0.703}^{\pm 0.007}$ | $\mathbf{0.399}^{\pm 0.000}$ | $\mathbf{0.555}^{\pm 0.005}$ | $\mathbf{0.638}^{\pm 0.004}$ |

Table 18: Abalation of modeling strategy on the Motion-X dataset (FID, TMR-R-Precision$^{(256)}$, and R-Precision$^{(32)}$ metrics).

As shown in Table 18, modeling body and hands separately will result in a large performance loss in whole-body motion generation. As a result, we take the H$^2$VQ and Hierarchical-GPT as the technical design choice.

## K  BROADER IMPACT AND LIMITATION

In this section, we will discuss the border impact and limitations.

**Broader Impact.** On the one hand, we explore the whole-body motion generation task and leverage the large-scale whole-body mocap dataset Motion-X to pre-train a motion-text-aligned prior. These could be a foundation for the field-related research community. Besides, based on the motion reconstruction via the proposed discrete latent compression scheme of human motions and large-scale motion data training, the pre-trained HumanTOMATO can provide motion prior, like VPoser Pavlakos et al. (2019). It can also benefit Motion Capture models (*e.g.*, OSX (Lin et al., 2023b)) denoising and improve the precision. On the other hand, expressive, text-controllable, and high-quality motion generation can be implemented for many practical application scenarios, such as motion generation for games and animations, robotics, and motion interaction.

**Limitation.** Although this work makes great progress on the novel task, and the significant improvement of motion reconstruction and text-aligned generation, it still has some shortcomings. First, the natural textual description utilization for whole-body motion generation needs to be further explored. This work simply uses the sequential semantic descriptions following previous works without frame-level or fine-grained whole-body descriptions. Second, the face generation lacks a unified generation scheme. Due to the limited holistic facial expression data and face motion descriptions (e.g., only commonly used emotion here), a simple condition VAE is not the best design choice. As rich data comes, a unified framework could be future work.

