# OpenReview forum: "HumanTOMATO: Text-aligned Whole-body Motion Generation"
_ICLR.cc/2024/Conference — Submitted to ICLR 2024_

### Official Review · Reviewer_yjZC · 2023-10-26

**Soundness:** 3 good
**Presentation:** 4 excellent
**Contribution:** 3 good
**Rating:** 8
**Confidence:** 3

**Summary:**

This paper proposed to solve the novel task of text-driven whole-body motion generation, which generates body, hand, and face motion simultaneously given a text description. The proposed method includes a holistic hierarchical VQ-VAE for hand and body motion encoding, a hierarchical GPT for hand and body motion generation, a cVAE for facial expression, and a pretrained text and motion alignment model. The paper also proposed to evaluate the alignment between text and motion with a novel evaluation metric TMR-R-Precision(256) and TMR-Matching Score. Experiments were conducted on the Motion-X, GRAB, and HumanML3D datasets.

**Strengths:**

- This is the first paper to generate holistic and vivid motions with body, hand, and facial expressions.
- The authors pretrained a text-motion retrieval to align the text and motion embedding, bypassing the semantic gap between CLIP-based text embedding and motion.
- The proposed methods show a clear advantage against SOTA motion generation methods in common metrics, except for multi-modality and diversity.

**Weaknesses:**

### Major issue

- The paper emphasized generating “vivid motions,” and the key for a motion to be vivid is to have vivid facial expressions. However, the proposed solution does not show promising results in facial expression generation. The facial expression part is not evaluated against any baseline method, and the facial cVAE is disconnected from the other parts of the proposed method.

### Minor issues

- In section 2.1, $F$ should be $L$?
- In section 2.2, the sentence is repeated twice

    > In the encoding phase, we input hand and body motions, yielding hand and body tokens through the hand encoder EncH(·) and the body encoder EncB(·),  respectively.
    >
- In Appendix C.1, Algorithm 1, line 4 in the for loop, $\hat{z}^B=..., \hat{z}^B));\mathcal{C}^B);$ the second $\hat{z}^B$ should be $z^B$.

**Questions:**

- What’s the reason behind the name hierarchical-GPT? The model seems to interleave instead of hierarchical to me.
- Why is it that in H2VQ, the hand is encoded before the body, and in Hierarchical-GPT, the body token is predicted before the hand? Is there any reason besides empirical performance advantages?
- To what extent can the method generalize to OOD text descriptions?
- In the supplementary video, in the T2M-GPT examples, the characters are all moving backward. Please confirm if the examples are rendered correctly.

---

> ### Author Response · Authors · 2023-11-19
> **Response to Reviewer yjZC**
>
> Dear Reviewer yjZC, thanks for your valuable comments and insightful suggestions. Please find our response to your comments below.
>
> - Q1: About facial generated results and comparison.
> - A1: Thanks for your suggestions.
>     - In the paper, it is hard to present the dynamics of face motions. We provide more generated face motion videos in the supplementary. Please check it. The analysis of face motions are discussed in Appendix F. As this is the first work targeting at face motion generation, we provide a simple baseline on text2face motion generation. Besides, the facial cVAE is not the main technical design, but a component to build up the whole system.
>     - On reason why model the face and other motions separately. The text2motion and text2face are both one-to-many mapping and the face motion is not highly coherent with other motions. Besides, the face motions are more highly related to the speeches [1, 2]. Therefore, we treat body and face motions as highly disentangled and model them respectively. This fashion of motion modeling is also been used in previous research [3]. Moreover, the face motion is represented in expression parameters, which is different from skeleton-based motions. This makes it hard to build an all-in-one system.
>
>     [1]: Richard, Alexander, et al. "Meshtalk: 3d face animation from speech using cross-modality disentanglement." *Proceedings of the IEEE/CVF International Conference on Computer Vision*. 2021.
>
>     [2]: Fan, Yingruo, et al. "Faceformer: Speech-driven 3d facial animation with transformers." *Proceedings of the IEEE/CVF Conference on Computer Vision and Pattern Recognition*. 2022.
>
>     [3]: Habibie, Ikhsanul, et al. "Learning speech-driven 3d conversational gestures from video." *Proceedings of the 21st ACM International Conference on Intelligent Virtual Agents*. 2021.
>
> ---
>
> - Q2: minor issues.
> - A2: We apologize for these issues. We have revised them in the revision. Please check it.
>
> ---
>
> - Q3: What’s the reason behind the name hierarchical-GPT? The model seems to interleave instead of hierarchical to me.
> - A3: Based on the Forward Kinematics (FK) and the skeleton tree, body motions can be treated as the next hierarchy of body motions. Based on these, we name it Hierarchical-GPT.
>
> ---
>
> - Q4: Why is it that in H$^2$VQ, the hand is encoded before the body, and in Hierarchical-GPT, the body token is predicted before the hand? Is there any reason besides empirical performance advantages?
> - A4: Similar to Q3, based on the FK and the skeleton tree, hand motions are highly dependent on body motions. This is the intuitive insight of this design.
>
> ---
>
> - Q5: To what extent can the method generalize to OOD text descriptions?
> - A5: We greatly appreciate your insightful suggestions to help us explore the boundaries of model performance. We provide some OOD text testing, whose results are shown in the supplementary (`supplementary/OOD/`).
>     - We test the robustness of different subject descriptions, like "the guy", "the woman".
>     - We also test the robustness of different tenses, like simple tense and continuous tense.
>     - We test some of the novel motion descriptions, which are few or do not appear in the dataset. Both success and failure cases are provided. **We find some cases interesting to share.** (1) For the "playing basketball" case, the generated results are about a man who is dunking. (2) For the "a man is sleeping" and "a man is sleeping on the ground" cases, sometimes it succeeds, sometimes it fails. Note that there are no cases describing the sleeping motion directly. The success case is similar to "the person is lying on the ground.", which we think is reasonable. The failure case is a swimming motion. We analyze this is because swimming and lying are similar motions. We also analyze the swimming and lying motions in the retrieval demo (`supplementary/retrieval-demo.mp4`).
>
> ---
>
> - Q6: In the supplementary video, in the T2M-GPT examples, the characters are all moving backward. Please confirm if the examples are rendered correctly.
> - A6: Thanks for the reminder. We carefully checked the visualization process of T2M-GPT. Everything goes well. We also provide more results of T2M-GPT generated results in the supplementary.

---

> > ### Comment · Reviewer_yjZC · 2023-11-22
> >
> > The authors' response have addressed my concerns.

---

> > > ### Author Response · Authors · 2023-11-22
> > > **Re: Official Comment by Reviewer yjZC**
> > >
> > > Dear Reviewer yjZC,
> > >
> > > Thanks for your insightful feedback. We are glad that our response has resolved your concern. Your comment is very valuable for us to improve our manuscript!
> > >
> > > Best,
> > >
> > > Authors

---

### Official Review · Reviewer_JrQp · 2023-10-31

**Soundness:** 3 good
**Presentation:** 3 good
**Contribution:** 2 fair
**Rating:** 6
**Confidence:** 3

**Summary:**

This paper focuses on the text-driven whole-body motions generation, including facial expressions, hand gestures, and body movements. The proposed framework consists of a holistic hierarchical VQ-VAE to compress the whole-body motion into two-level discrete codes. It also features a hierarchical-GPT model that predicts the discrete motion codes from input textual descriptions in an auto-regressive manner. Additionally, the author proposes a pre-trained text-motion-alignment model to enhance the alignment between given text and generated motions.

**Strengths:**

1. The paper is the first to target the text-driven whole-body motions generation task, aiming at generating high-quality, diverse, and coherent facial expressions, hand gestures, and body motions.

2. The paper introduces a novel framework for text-driven whole-body motion generation, featuring a Holistic Hierarchical Vector Quantization for learning informative and compact representations at low bit rates, along with a Hierarchical-GPT for predicting hierarchical discrete codes for body and hand motions in an autoregressive manner.

3. The paper proposes a pre-trained text-motion alignment model, which serves to provide textual embeddings instead of commonly used CLIP embeddings. Furthermore, it offers motion-text alignment supervision during the training process.

**Weaknesses:**

In my view, the proposed H2VQ and Hierarchical-GPT just extend the model introduced in T2M-GPT by incorporating hand gesture modeling. These modifications are rather straightforward. Firstly, in the context of vector quantization, they integrate the hand pose vector quantization with the body pose using a hierarchical strategy rather than directly quantizing the whole body pose. Secondly, The T2M-GPT has been modified to decode the body pose code and hand pose code alternately, rather than directly outputting the whole body pose code. The utilization of TMR for encoding textual descriptions is a more intelligent choice compared to CLIP embedding, and the incorporation of motion-text alignment supervision appears to be beneficial during the training. However, it's important to note that text-motion alignment has been employed in various previous works, including TEMOS, and the proposed TMR merely adopts a contrastive learning way through a retrieval target. The paper presents a substantial amount of contributions, but the technical designs lack novelty and a certain level of appeal from my perspective. As a result, I would recommend a rating of marginally above the acceptance.

**Questions:**

1. Currently, is it feasible or essential to generate diverse and realistic human poses and facial expressions using the available datasets? To my knowledge, most of the existing datasets lack diversity and realism in hand poses and facial expressions. From visualization results, I can discern certain minor distinctions in hand poses, although they may not be highly realistic, and I cannot find the differences in the generated facial expressions.

2. How about the comparison with a simple baseline that directly combines SOTA models for facial expression, hand pose, and body pose generation?

Minor Fix:

1. On page 3, where it mentions, "where $F$ and $d$ denote the number of frames and the dimension ...", it may be advisable to replace $F$ with $L$ to maintain consistent notation.

2. On page 4, there is a repetition of the sentence: "In the encoding phase, we input hand and body motions, ...".

---

> ### Author Response · Authors · 2023-11-19
> **Response to Reviewer JrQp (Part 1/2)**
>
> Dear reviewer JrQp, we sincerely acknowledge your insightful suggestions. Please find our response to your comments below.
>
> - Q1: About technical novelty and contribution.
> - A1: We sincerely acknowledge the discussion about the novelty of our work. We would like to clarify as follows.
>     - For the VQ part, we explore how to improve the generation quality efficiently by enhancing reconstruction precision. However, increasing the codebook size is almost in vain. Therefore, we propose the H$^2$VQ to replace the traditional VQ, which benefits the whole-body motion generation task significantly (See Table 2 and Appendix G). Moreover, to verify our hierarchical modeling insight, we adapt our H$^2$VQ as RVQ to the body-only motion generation. The results verify that this modeling strategy also benefits body-only motion generation.
>     - For the Hierarchical-GPT contribution, as there are two (hierarchical) codebooks in the H$^2$VQ, it is also necessary to design the corresponding hierarchical token prediction mechanism.
>     - For the TMR, our core contribution is not how to train a TMR, but how to use a TMR. Since April 2022, MotionCLIP has introduced the CLIP to generate motion. After, a lot of follow-up research (like MDM, MotionDiffuse, T2M-GPT, and MLD) took the CLIP as a text encoder and regarded it as a default setting. **However, the CLIP cannot include any dynamic clues of motion as it is aligned with images. The text2motion community seems not to realize the problem with using CLIP. We revisit the problem in this work and well discuss how to use TMR.** The results of our method benefit the motion generation and text-motion alignment significantly (see Table 3). We believe this will provide some new understanding to the community.
>
>     We claim these are something valuable to peers and the community, which we think is novel [1].
>
>     [1]: Micahel J. Black, Novelty in Science. https://perceiving-systems.blog/en/news/novelty-in-science.
>
> ---
>
> - Q2: About the TMR v.s. TEMOS.
> - A2: TEMOS only learns the alignment between positive pairs. However, TMR also learns the misalignment between positive pairs. TEMOS is not good enough for alignment. To verify the good alignment of TMR, we compare its retrieval ability with TEMOS As shown in the table the TMR enjoys a good alignment between texts and motions by the contrastive training objective, which makes it with a larger margin than TEMOS in retrieval. This good retrieval ability provides a better alignment of two modalities, and provide a better motion-text alignment for motion generation. Details are shown in Appendix D.3.
>
>
>     | Protocol | R@1 (T2M) | R@2 (T2M) | R@3 (T2M) | R@5 (T2M) | R@10 (T2M) | R@1 (M2T) | R@2 (M2T) | R@3 (M2T) | R@5 (M2T) | R@10 (M2T) |
>     | --- | --- | --- | --- | --- | --- | --- | --- | --- | --- | --- |
>     | Protocol A (TEMOS) | 0.034 | 0.062 | 0.082 | 0.117 | 0.178 | 0.034 | 0.067 | 0.096 | 0.137 | 0.204 |
>     | Protocol A (TMR) | 0.051 | 0.098 | 0.131 | 0.192 | 0.301 | 0.066 | 0.118 | 0.163 | 0.233 | 0.350 |
>     | Protocol B (TEMOS) | 0.085 | 0.143 | 0.190 | 0.237 | 0.308 | 0.112 | 0.135 | 0.155 | 0.190 | 0.246 |
>     | Protocol B (TMR) | 0.089 | 0.152 | 0.194 | 0.273 | 0.401 | 0.169 | 0.205 | 0.238 | 0.298 | 0.395 |
>     | Protocol C (TEMOS) | 0.233 | 0.341 | 0.412 | 0.492 | 0.601 | 0.263 | 0.360 | 0.426 | 0.504 | 0.614 |
>     | Protocol C (TMR) | 0.445 | 0.609 | 0.700 | 0.799 | 0.883 | 0.407 | 0.578 | 0.673 | 0.795 | 0.883 |
>     | Protocol D (TEMOS) | 0.502 | 0.641 | 0.717 | 0.802 | 0.897 | 0.528 | 0.654 | 0.725 | 0.807 | 0.907 |
>     | Protocol D (TMR) | 0.716 | 0.854 | 0.907 | 0.946 | 0.977 | 0.771 | 0.893 | 0.938 | 0.968 | 0.985 |
>
> ---
>
> - Q3: Is it feasible or essential to generate diverse and realistic human poses and facial expressions using the available datasets?
> - A3: We use the Motion-X dataset, which has been peer-reviewed and accepted by NeurIPS-2023 dataset and benchmark track. The dataset is the largest existing text-motion dataset. We take the first attempt to tackle the whole-body motion generation task based on this dataset.
>
> ---
>
> - Q4: I cannot find the differences in the generated facial expressions.
> - A4: Thanks for your suggestions. In the paper, it is hard to present the dynamics of face motions. We provide more generated face motion videos in the supplementary. Please check it.

---

> ### Author Response · Authors · 2023-11-19
> **Response to Reviewer JrQp (Part 2/2)**
>
> - Q5: How about the comparison with a simple baseline that directly combines SOTA models for facial expression, hand pose, and body pose generation?
> - A5: Thanks for the suggestions. If hand and body motions are generated separately, it will result in the incoherence of motions. We provide an ablation on whether to model them separately in the following table. Both hand and body motions are generated by T2M-GPT. The results show that modeling body and hands separately will result in a large performance loss in whole-body motion generation. We also update this in Appendix J.
>
>
>     |  | FID | TMR_top1 (256) | TMR_top2_RP (256) | TMR_top3_RP (256) | top1_RP (32) | top2_RP (32) | top3_RP (32) |
>     | --- | --- | --- | --- | --- | --- | --- | --- |
>     | separately | 2.209±0.047 | 0.306±0.003 | 0.459±0.002 | 0.552±0.002 | 0.359±0.002 | 0.551± 0.003 | 0.666±0.002 |
>     | Ours | **1.174±0.015** | **0.399±0.000** | **0.555±0.005** | **0.638±0.004** | **0.416±0.009**  | **0.603±0.007**  | **0.703±0.007** |
>
> ---
>
> - Q6: Minor Fix.
> - A6: We apologize for these issues. We have revised them in the revision. Please check it.

---

> ### Comment · Reviewer_JrQp · 2023-11-22
> **Thanks for your reply.**
>
> I have no further questions.

---

> > ### Author Response · Authors · 2023-11-22
> > **Re:Thanks for your reply.**
> >
> > Dear Reviewer JrQp:
> >
> > Thanks for your suggestions and your valuable feedback. Your comment is very meaningful to further improve our manuscript!
> >
> > Best,
> >
> > Authors

---

### Official Review · Reviewer_bynW · 2023-10-31

**Soundness:** 3 good
**Presentation:** 2 fair
**Contribution:** 2 fair
**Rating:** 5
**Confidence:** 4

**Summary:**

This paper proposed a framework for whole-body motion generation from text. It includes several core designs: 1) a holistic hierarchical VQ-VAE based on RVQ for body and hand motion reconstruction; 2) a hierarchical GPT for predicting fine-grained body and hand motions; 3) a pre-trained text-motion-alignment model used as a prior for text-motion generation stage explicitly; 4) a text-motion alignment supervision in the GPT preditor. Comprehensive experiments verify that the proposed model has significant advantages both quantitively and qualitatively.

**Strengths:**

The authors pioneered the task of whole-body motion (including the face and hand motion) generation from speech. To generate fine-grained hand and face motions, two core technique designs were introduced: 1) a holistic hierarchical VQ-VAE based on RVQ for body and hand motion reconstruction; and 2) a hierarchical GPT-based generation. To achieve a good alignment between text and motion, a text-motion retrieval model is pre-trained and used as a prior for the text-motion generation stage explicitly. Extensive quantitative and qualitative experiments were conducted to demonstrate the efficiency of the proposed method.

**Weaknesses:**

1. The technical contribution of this paper seems somewhat limited in the following:
(1) the pipeline of the method is similar to the T2M-GPT where a VQ-VAE is used for motion reconstruction and a transformer-based GPT model is used for motion generation, while the tasks are different;
(2) the pretraining of a motion encoder and a text encoder via aligning text and motion in a contrastive way is also not new such as TMR, and further using it to replace the clip is natural in the prediction stage.
2. From the example of the visualizations, the textual description for facial motions focused on emotion (like happily, angrily), but the generated face shown in the paper is static, rather than dynamic motions, which lacks the demonstration of the emotion dynamics.

**Questions:**

Except for the above weaknesses, there are a few questions as follows:
1. In Table 2, about the H2VQ, what is the size of the codebooks for reconstructing the body and hand motion? Besides, for vanilla VQ-VAE and RVQ, do you separately model the hand and body motions and then combine the motion as body-hand motions? How is the performance for VQ-VAE and RVQ when increasing from 1024 to 4096?
2. Regarding facial generation,  what the text embedding is used? Clip or TMR?
3. Since the hand & body, and face motions are separately modeled, how is the coherence of the generated motions?

---

> ### Author Response · Authors · 2023-11-19
> **Response to Reviewer bynW (Part 1/2)**
>
> Dear reviewer bynW, we sincerely acknowledge your suggestions. We begin to answer the questions and concerns by clarification.
>
> **Clarification**: For "The authors pioneered the task of whole-body motion (including the face and hand motion) generation from speech.", we sincerely thank your appreciation for our pioneered contribution. **However**, this work is not related to **speech** driven motion generation, but **text** driven motion generation.
>
> - Q1: About technical contributions.
> - A1: Thanks for discussing our technical contributions. We would like to discuss them from two aspects.
>     - About `similar to T2M-GPT` concern. The two-stage training pipeline (VQ-VAE + GPT prediction) in T2M-GPT is almost nothing new. The training pipeline has been widely explored by the vision community (Taming Transformer [1], Video GPT[2], DALL-E-1 [3]). T2M-GPT contributes to the motion community by introducing this fashion to motion generation and verifying its effectiveness. Therefore, we take this effective fashion as a base to generate motion. Our core contribution is (1) to propose a more effective VQ for whole-body motion generation; and (2) to explore how to use language prior properly and which kind of language prior is more proper. We think these two contributions will help the community to generate human motions.
>     - For the H$^2$VQ design, we carefully designed it for our novel text-to-whole-body-motion task with hierarchical body-hand modeling, which reflects our key insights on the task.
>     - For the TMR part, in April 2022, MotionCLIP introduced the CLIP to generate motion. After that, a lot of follow-up research (like MDM, MotionDiffuse, T2M-GPT, and MLD) took the CLIP as a text encoder and regarded it as a default setting. In this work, we revisit how to introduce better language prior to motion generation. We think our answer on **how to use** T-M-aligned language prior is essential to the community for the first time.
>
> [1]: Esser et.al. Taming transformers for high-resolution image synthesis. *CVPR,* 2021.
>
> [2]: Yan et.al. VideoGPT: Video Generation using VQ-VAE and Transformers. 2021.
>
> [3]: Ramesh et.a. Zero-Shot Text-to-Image Generation. ICML, 2021.
>
> ---
>
> - Q2: About facial motion dynamics.
> - A2: Thanks for your suggestions. In the paper, it is hard to present the dynamics of face motions. We provide more generated face motion videos in the supplementary. Please check it.
>
> ---
>
> - Q3: About the H$^2$VQ, what is the size of the codebooks for reconstructing the body and hand motion?
> - A3: The codebook size of $Q_1$ and $Q_2$ are both 512.
>
> ---
>
> - Q4: For vanilla VQ-VAE and RVQ, do you separately model the hand and body motions and then combine the motion as body-hand motions?
> - A4: No. For vanilla VQ and RVQ, we take the body-hand motions as a whole during training. We provide an ablation on whether to model them separately in the following table. We compare the H$^2$VQ with 2 separate VQs. The generated results show that modeling body and hands separately will result in a large performance loss in whole-body motion generation. We also update this in Appendix J.
>
>
>     |  | FID | TMR_top1 (256) | TMR_top2_RP (256) | TMR_top3_RP (256) | top1_RP (32) | top2_RP (32) | top3_RP (32) |
>     | --- | --- | --- | --- | --- | --- | --- | --- |
>     | separately | 2.209±0.047 | 0.306±0.003 | 0.459±0.002 | 0.552±0.002 | 0.359±0.002 | 0.551± 0.003 | 0.666±0.002 |
>     | Ours | **1.174±0.015** | **0.399±0.000** | **0.555±0.005**  | **0.638±0.004** | **0.416±0.009**  | **0.603±0.007**  | **0.703±0.007** |
>
> ---
>
> - Q5: How is the performance for VQ-VAE and RVQ when increasing from 1024 to 4096?
> - A5: We conduct the experiments on the size of VQ, RVQ, and H$^2$VQ. Results are shown in the following table (on MPJPE metric). As shown in the table, the scaling of the codebook size is almost in vain. This observation supports the basic intuition on the designing of H$^2$VQ. We provide a detailed explanation in Appendix G.2.
>
>
>     | Size | 512 | 1024 | 4096 |
>     | --- | --- | --- | --- |
>     | VQ | 140.7 | 139.3 | 134.2 |
>     | RVQ | 110.9 | 111.2 | 116.8 |
>     | H$^2$VQ | 93.0 | 83.9 | 84.9 |

---

> ### Author Response · Authors · 2023-11-19
> **Response to Reviewer bynW (Part 2/2)**
>
> - Q6: Regarding facial generation, what the text embedding is used? Clip or TMR?
> - A6: None of both. We discuss the model in Appendix B.4 and the text encoder comes up with a pre-trained DistillBERT and a 6-layer transformer.
>
> ---
>
> - Q7: Since the hand & body, and face motions are separately modeled, how is the coherence of the generated motions?
> - A7: We acknowledge that the coherence issue. The text2motion is a kind of one-to-many mapping and the face motion is not highly coherent with other motions. Besides, the face motions is more highly related to the speeches [1, 2]. Therefore, we treat them as highly disentangled and model them respectively. This fashion of motion modeling is also been used in previous research [3]. Moreover, the face motion is represented in expression parameters, which is different from skeleton-based motions. This makes it hard to build an all-in-one system.
>
>
>     [1]: Richard, Alexander, et al. "Meshtalk: 3d face animation from speech using cross-modality disentanglement." *Proceedings of the IEEE/CVF International Conference on Computer Vision*. 2021.
>
>     [2]: Fan, Yingruo, et al. "Faceformer: Speech-driven 3d facial animation with transformers." *Proceedings of the IEEE/CVF Conference on Computer Vision and Pattern Recognition*. 2022.
>
>     [3]: Habibie, Ikhsanul, et al. "Learning speech-driven 3d conversational gestures from video." *Proceedings of the 21st ACM International Conference on Intelligent Virtual Agents*. 2021.

---

> ### Author Response · Authors · 2023-11-22
> **Further discussion with Reviewer bynW**
>
> Dear Reviewer bynW:
>
> Thanks again for your efforts and suggestions for this submission. The deadline for the author-reviewer discussion is approaching. To enhance this paper, we authors hope you can check our response and confirm whether there are unclear explanations. We want to solve them for you.
>
> Best,
>
> Authors

---

### Official Review · Reviewer_MYWU · 2023-11-05

**Soundness:** 3 good
**Presentation:** 4 excellent
**Contribution:** 2 fair
**Rating:** 5
**Confidence:** 4

**Summary:**

This paper is the first work that can generate whole-body motion from text description. This paper first proposes a Holistic Hierarchical Vector Quantization (H$^2$VQ) scheme to model the correlation between body and hand. Authors notice that facial expression is largely independent of body and hand, so they train a conditional VAE to generate facial motion independently. This scheme is reasonable, which is also verified in the task of SMPL-X reconstruction. Compared with the H$^2$VQ and facial cVAE, the text-to-motion alignment module is more interesting. If the author can release this module as claimed in their manuscript, this module will bring some meaningful progress to the community of text2motion.

**Strengths:**

The authors propose a promising task and give a thoughtful solution. From the results, we can easily judge the effectiveness of the proposed method. The writing is also very fluent.

**Weaknesses:**

1. The title '3.1.2 Evaluation' somehow is easily misleading. In this section, you introduced the evaluation metrics and compared methods. How about changing to 'Evaluation Details'?
 2. I feel that the focus of this article is on how to generate physical movements in the hands. The discussion about facial cVAE is limited, and I have not found any experiments to analyze this module.
3. Although it's hard to find compared methods in text-aligned whole-body motion generation, the authors can compare solely body generation results with previous works. But I didn't find this part. There are too few methods for comparison.

**Questions:**

My questions have been listed in the above Weaknesses. I have one more question on this paper: From visual results, I noticed some physically implausible artifacts, such as foot sliding. Can you give some discussion on this point?

---

> ### Author Response · Authors · 2023-11-19
> **Response to Reviewer MYWU**
>
> We sincerely appreciate your review of our contribution to the community and acknowledge for helping us polish this work. We promise to make all codes public if accepted. Additionally, wishing our TMR would play a similar role in the motion generation community like the CLIP in the vision-language area, we would release the training codes, pre-trained models, and retrieval demos if accepted.
>
> - Q1: How about changing into 'Evaluation Details'?
> - A1: We acknowledge your suggestion. We change the ‘Evaluation’ to ‘Evaluation Details’. Please check the revision of the manuscript.
> ---
> - Q2: discussion about facial cVAE.
> - A2: We provide more visualization results in the supplementary, where some success cases and failure cases are well discussed. Due to the space limitation, we discuss the facial cVAE in Appendix F.1 (Table 9).
> ---
> - Q3: compare solely body generation results with previous works. There are too few methods for comparison.
> - A3: Our method is carefully designed for whole-body motion generation. For comparison, we compare with previous body-only motion generation methods by replacing the body motion with body-hand motions. That is to say, we replace the body pose in each frame with the body-hand pose directly. We think this is the most reasonable way for comparison. For concerns about the number of baselines, we compare the HumanTOMATO with two more baselines (TEMOS and MotionDiffuse). The results in the following table show the good generation ability of HumanTOMATO.
>
>
>     | Methods | FID | R-Precision Top1 | R-Precision Top2 | R-Precision Top3 | Matching_score | Diversity |
>     | --- | --- | --- | --- | --- | --- | --- |
>     | TEMOS | 9.147 | 0.279 | 0.442 | 0.555 | 5.482 | 9.764 |
>     | MotionDiffuse | **1.130** | 0.391 | 0.587 | 0.695 | 3.950 | 10.580 |
>     | Ours | 1.174 | **0.416** | **0.603** | **0.703** | **3.894** | **10.812** |
>
> [1]: Zhang, Mingyuan, et al. "Motiondiffuse: Text-driven human motion generation with diffusion model." *arXiv preprint arXiv:2208.15001* (2022).
>
> [2]: Petrovich, Mathis, Michael J. Black, and Gül Varol. "TEMOS: Generating diverse human motions from textual descriptions." *European Conference on Computer Vision*. Cham: Springer Nature Switzerland, 2022.
>
> ---
> - Q4: Discussion on physically implausible artifact.
> - A4: Thanks for the discussion on the physically implausible artifact. This mainly comes from three aspects.
>     - Firstly, most text2motion efforts proposed by the community are hard to generate physically plausible motions [1]. Generating physically plausible motions requires post-processing in a simulation environment.
>     - Secondly, whole-body motion generation is much more challenging to generate than previous body-only motion generation. The number of joints scales from 24 to 52, making generation more challenging. Notably, adapting the motion from body-only motion to whole-body motion following the motion representation setting in [2] also makes the model hard to learn due to the high-dimensional data. Therefore, we even explore which kind of motion representation is good for whole-body motion generation. The results are discussed in Appendix B.1, which shows the representation w/o rotations is a more efficient and effective motion representation. The exploration of motion representation might be helpful to the community and it is our efforts to explore how to generate realistic motions.
>
> [1]: Yuan et.al., Physdiff: Physics-guided human motion diffusion model. ICCV, 2023
>
> [2]: Guo et.al., Generating diverse and natural 3d human motions from text. CVPR, 2022.

---

> ### Comment · Reviewer_MYWU · 2023-11-22
> **Further questions on facial expression**
>
> Dear authors,
>
> Thanks for your response!
> I have some questions about the qualitative results about hand and facial expressions:
>
> Based on the results of existing images, I can determine that the body movements match the text description, but it's hard to judge the hand movements and facial expressions. Can you provide some more obvious examples to highlight the actions of the hands and facial expressions? For example, making an OK gesture with the hands. Regarding facial expressions, my issue is that you only provided descriptions like 'happy', 'sad', 'bothered'. From Fig. 9 in the supplemental material, it feels like the expressions in all the movements are the same. For instance, the result corresponding to the description "sport fitness standing left and right leg swing, happily".
>
> I noticed that Reviewer bynW also raised the same question on the dynamics of facial expression. After watching videos in the Supp, the question still exists. From the 4-minute mark to the end of the video, I didn't see any obvious dynamics of the hands and face. From the results in the folder of `expressive cases', I can see the facial dynamics. But there comes a question: why can't the results of the paper and the video in the Supp show obvious facial expressions, even if I zoom in?
>
> Minor: it would be better to add a readme in the Supp for more clear clarity.

---

> ### Author Response · Authors · 2023-11-22
> **Re: Further questions on facial expression**
>
> Dear reviewer MYWU,
>
> Thanks for your suggestions and your valuable feedback. We answer your questions as follows.
>
> Q1: Can you provide some more obvious examples to highlight the actions of the hands and facial expressions?
>
> A1: We provide some expressive cases of `supplementary/visual_result/make_ok_gesture_disgusted.mp4` and  `supplementary/visual_result/standing_during_hissing_angrily.mp4`. For the case “make ok gesture disgusted”, we can see the man is frowning. For the case “standing during hissing angrily”, it seems the man is teaching someone a lesson.
>
> ---
>
> Q2: Regarding facial expressions, my issue is that you only provided descriptions like 'happy', 'sad', 'bothered'.
>
> A2: Thanks for your suggestions. Except for these cases, we also provide some expressions like depressly and concentrating (`supplementary/face/expressive cases`). Besides, in our daily life, human face motions are mostly neutral. Therefore, we provide more neutral cases in supplementary (`supplementary/face/netural expression cases`).
>
> ---
>
> Q3: From the 4-minute mark to the end of the video, I didn't see any obvious dynamics of the hands and face.
>
> A3: Thanks for this question. We answer this question from two aspects.
>
> - For hands part, please check the 4'25''-4'35'' part. We find something interesting on the hand motions. The generated results of HumanTOMATO are **even more vivid than GT**. When scratching his face, the man even stretch out his index finger on face. In the motion, the man changes the hand motions from making a fist to scratching the face. We also highlight the hand pose with the red circle in 4'35''. Besides, we even discuss the fine-grained hand motions from 5'07'' to 5'18'', which shows the good performance of our model against previous work. From 5'50'' to 6'30'', we conduct the experiments on body-only dataset (HumanML3D, without hand and face) to show our design also benefits the body-only motion generation.
> - For the face part, from the 4-minute mark to the end of the video, we mainly discuss **the comparison with baselines and ablations (H$2$VQ and TMR)**. However, previous methods cannot generate face motions. As a result, we only generate body-hand motions for comparison and set the expression of face as neutral by default. We are glad our response can provide facial dynamics recognized by you. The generated dynamic face motions can be easily concatenated to the body.
>
> ---
>
> Q4: For the expressive face motion concern. Why can't the results of the paper and the video in the Supp show obvious facial expressions, even if I zoom in?
>
> A4: For the “it feels like the expressions in all the movements are the same” comment, we found the expressions in the Figure 9 are not the same. For instance, the “sport fitness standing left and right leg swing, happily.” and “Play Big ruan, sadly.” reflect the happy and sad expressions **when you enlarge the figure**. For the case “sport fitness standing left and right leg swing, happily.”, we can clear find the person grins. Besides, in Figure 10, “a person walks slowly in a half circle counterclockwise while holding something, in disgust.” also shows the corresponding expression of “in disgust”. In our submissions, we can clearly see expressions in Figure 9 and 10.
>
> ---
>
> We additionally added the README in the supplementary.
>
> Best,
>
> Authors

---

> > ### Author Response · Authors · 2023-11-23
> > **To Reviewer MYWU**
> >
> > Dear Reviewer MYWU,
> >
> > Thanks again for your efforts and suggestions for this submission. The deadline for the author-reviewer discussion is approaching (only several hours). We further answered your latest questions. We want to confirm whether there are unclear explanations and descriptions here. We could further clarify them.
> >
> > Best,
> >
> > Authors

---

### Author Response · Authors · 2023-11-19
**General Response**

Dear all reviewers, AC, and SAC,

Thanks for handling this submission. We would like to thank all reviewers for their insightful suggestions and valuable efforts to polish this submission. We find it encouraging to see that our work is **contributive** (MYWU, yjZC), **pioneering** (MYWU, bynW, JrQp, yjZC), and **written fluently** (MYWU). We are also delighted with the reviews’ recognition of **competitive** results (MYWU, bynW, yjZC).

We revised the manuscript according to the suggestions from reviewers, highlighted in blue. Besides, we update the supplementary. We summarized the major changes as:

- Fix subsection titles, typos, and repeating sentences.
- Provide ablation on the size of VQ, RVQ, and H$^2$VQ. (Appendix G.2)
- Provide ablation on whether to model different parts of motions separately. (Appendix J)
- Update the demo video. (`supplementary/demo.mp4`)
- Provide the web demo screen recording of the TMR retrieval. (`supplementary/ret-demo.mp4`)
- Provide the comparison between TMR and TEMOS. (Appendix D.3)
- Provide more test-in-the-wild cases. (`supplementary/OOD/`)
- Provide more generated results of T2M-GPT. (`supplementary/T2M-GPT/`)
- Provide more results analysis on the visualization of face motions. (`supplementary/face/`)

For concerns on the novelty and contribution, we provide re-clarification on the novelty and contribution. Our contribution comes from the following aspects.

1. A novel vector quantization method for whole-body motion.
    - We carefully investigate whether the VQ-VAE can enjoy the codebook scaling on motion generation. Therefore, we propose the H$^2$VQ as a new quantization technique for whole-body motion generation. The detailed comparison is detailed in Appendix G.1.
    - Based on this hierarchical modeling method, we additionally propose the Hierarchical-GPT to perform the latent code prediction in H$^2$VQ.

    These components build up the whole hierarchical motion generation system, which shows our key insight on generating high-quality whole-body motion.

2. A critical perspective on how to use language prior to the T2M task.
    - In this work, we revisit how to introduce better language prior for motion generation. In April 2022, MotionCLIP introduced the CLIP to generate motion. After that, a lot of follow-up research (like MDM, MotionDiffuse, T2M-GPT, and MLD) takes the CLIP as a text encoder and regards it as a default setting. However, the CLIP cannot include any dynamic clues of motion as it is aligned with images. **The text2motion community seems not to realize the problem with using CLIP. We revisit the problem in this work and well discuss how to use TMR.** And we think our answer on **how to use** T-M-aligned language prior is essential to the community for the first time.
    - We admit that the TMR part is not particularly technically difficult. But for more than a year, the entire community did not realize the importance of T-M alignment for motion generation. We formally explore this issue for the first time to promote the development of this task.
    - We added the TMR demo in the supplementary, which will be public. We hope TMR will play a role as CLIP in the motion generation community.

We claim these are something valuable to peers and the community, which we think is novel [1].

[1]: Micahel J. Black, Novelty in Science. https://perceiving-systems.blog/en/news/novelty-in-science.

---

> ### Comment · Reviewer_bynW · 2023-11-22
>
> Thanks for your response, which addressed most of my concerns.
>
> There are still some concerns: 1. As the authors respond, the body-hand are modeled jointly, what is about face motion modeling? Is it separated from the body-hand motion modeling? 2. As mentioned in the questions of the comments, how is the coherence of the generated motions if the face is separately modeled? 3. Since the task is generating holistic body motion, that is facial expression, body gesture, and hand pose. From the paper and supplemental material, the variety or nuance in the generated facial movements appears to be a bit lacking.

---

> > ### Author Response · Authors · 2023-11-22
> > **Re: Official Comment by Reviewer bynW**
> >
> > Dear reviewer bynW,
> >
> > We are glad that our response has resolved most of your concerns. We hope that in light of the response, you could consider improving the score. Our new response is as follows.
> >
> > We treat the 1 and 2 are similar questions and our answer is as follows.
> >
> > Q1 & Q2: As the authors respond, the body-hand are modeled jointly, what is about face motion modeling? Is it separated from the body-hand motion modeling? How is the coherence of the generated motions if the face is separately modeled?
> >
> > A1 & A2: Yes, they are modeled separately. This will not result in the incoherence of whole-body motions. The reasons are as follows.
> >
> > - The text2motion is a kind of one-to-many mapping and the face motion are not highly coherent with body-hand motions [1, 2]. Besides, the face motions is **more highly related to the speeches** [1, 2]. Therefore, we treat them (face and body-hand) as highly disentangled and model them respectively.
> > - This fashion of motion separated modeling is also been used in previous research [3] and will not result in incoherence.
> > - Moreover, the face motion is represented in expression parameters, which is quite different from skeleton-based motions. This makes it very hard to build an all-in-one system.
> >
> > [1]: Richard, Alexander, et al. "Meshtalk: 3d face animation from speech using cross-modality disentanglement." *Proceedings of the IEEE/CVF International Conference on Computer Vision*. 2021.
> >
> > [2]: Fan, Yingruo, et al. "Faceformer: Speech-driven 3d facial animation with transformers." *Proceedings of the IEEE/CVF Conference on Computer Vision and Pattern Recognition*. 2022.
> >
> > [3]: Habibie, Ikhsanul, et al. "Learning speech-driven 3d conversational gestures from video." *Proceedings of the 21st ACM International Conference on Intelligent Virtual Agents*. 2021.
> >
> > ---
> >
> > Q3: Since the task is generating holistic body motion, that is facial expression, body gesture, and hand pose. From the paper and supplemental material, the variety or nuance in the generated facial movements appears to be a bit lacking.
> >
> > A3: Thanks for your suggestions. The detailed path of updated face motions are in `supplementary/face/`.
> >
> > - In `supplementary/face/expressive cases`, we provide some expressive cases for face, like depressly, concentratingly, suppressingly and so on.
> > - In most scenarios of our daily life, as face motions are mostly neutral expressions, in  `supplementary/face/netural expression cases` , we provide some neutral cases for face motions.
> > - As this is the first attempt to generate whole-body motions, we also leave some failure cases in `supplementary/face/failure cases` frankly. Moreover, we additionally provide some expressive cases of `supplementary/visual_result/make_ok_gesture_disgusted.mp4` and  `supplementary/visual_result/standing_during_hissing_angrily.mp4`. For the case “make ok gesture disgusted”, we can see the man is frowning. For the case “standing during hissing angrily”, it seems the man is teaching someone a lesson.
> >
> > Best,
> >
> > Authors

---

### Author Response · Authors · 2023-11-22
**Further discussion**

Dear Reviewers:

Thanks again for your efforts and suggestions for this submission. The deadline for the author-reviewer discussion is approaching. To enhance this paper, we authors hope reviewers can check our response and confirm whether there are unclear explanations. We want to solve them for you.

Best,

Authors

---

### Public Comment · ~Zoltan_Adam_Milacski1 · 2023-12-04
**small error in paper**

Dear Authors,
This paper mentions "MotionCLIP (Tevet et al., 2022) renders the generated motions as images and then supervises the alignment between text and rendered images with the CLIP model". This is NOT true. MotionCLIP does NOT do differentiable rendering. Instead, it collects triplets of (motion, text, prerendered image) and uses CLIP similarity losses to match their embeddings. This small error does not change the story of your paper, but I think the text and Fig 3 a) should be edited accordingly. Additionally, AvatarCLIP is another paper worth citing.

---

### Meta-Review · Area_Chair_YrrA · 2023-12-09

**Metareview:**

This paper has mixed and mostly borderline reviews (2 borderline rejects, 1 borderline accept, 1 accept).  The reviewers appreciate the holistic approach taken on motion generation, tackling, not only the body but also the hands and face.  That having been said, several of the reviews raise concerns regarding the technical contribution and effectiveness of the approach, especially with regard to the hands and face, which is one of the key claims.

The AC has read through the reviews and author responses carefully.  After watching the videos, the AC agrees with the reviewers that the qualitative results are not indicative that the approach is fully effective.

**Justification For Why Not Higher Score:**

Questionable effectiveness.

**Justification For Why Not Lower Score:**

N/A

---

### Decision · Program_Chairs · 2024-01-16

Reject